# Optimal Rates in Continual Linear Regression via Increasing Regularization

Ran Levinstein*[†]    Amit Attia*[‡]    Matan Schliserman*[‡]    Uri Sherman*[‡]

Tomer Koren[§]    Daniel Soudry[¶]    Itay Evron[¶]

## Abstract

We study realizable continual linear regression under random task orderings, a common setting for developing continual learning theory. In this setup, the worst-case expected loss after $k$ learning iterations admits a lower bound of $\Omega(1/k)$. However, prior work using an unregularized scheme has only established an upper bound of $\mathcal{O}(1/k^{1/4})$, leaving a significant gap. Our paper proves that this gap can be narrowed, or even closed, using two frequently used regularization schemes: (1) explicit isotropic $\ell_2$ regularization, and (2) implicit regularization via finite step budgets. We show that these approaches, which are used in practice to mitigate forgetting, reduce to stochastic gradient descent (SGD) on carefully defined surrogate losses. Through this lens, we identify a fixed regularization strength that yields a near-optimal rate of $\mathcal{O}(\log k/k)$. Moreover, formalizing and analyzing a generalized variant of SGD for time-varying functions, we derive an *increasing* regularization strength schedule that provably achieves an optimal rate of $\mathcal{O}(1/k)$. This suggests that schedules that increase the regularization coefficient or decrease the number of steps per task are beneficial, at least in the worst case.

## 1 Introduction

In continual learning, a learner encounters a sequence of tasks and aims to acquire new knowledge without "forgetting" what was learned in earlier tasks. Many algorithmic approaches have been proposed to address this challenge [see surveys in 45, 42]. However, a deeper theoretical understanding is still needed to clarify the principles governing continual learning and is essential for the practical and reliable deployment of such methods.

We study standard regularization-based schemes in a setting with random task orderings. Both the setting and—especially—the schemes play a central role in the practical and theoretical continual learning literature, as discussed below. We find this combination mutually beneficial: (1) regularization improves the best known upper bound under random orderings, achieving an *optimal* rate; and (2) randomness facilitates analysis that motivates heuristics for setting the regularization strength.

We focus on two forms of regularization: a well-known *explicit* isotropic $\ell_2$ regularization, and *implicit* regularization induced by a finite number of gradient steps on the unregularized loss of each task. Prior work studied such schemes in restricted settings—*i.e.,* two tasks [28, 29], simplified data models [28, 48, 29], weak regularization [13, 23], or cyclic orderings [5]. In contrast, we consider *any* number of *regression* tasks drawn from *any* collection, under *random* orderings.

---

*Equal contribution.

[†]Department of Computer Science, Technion.

[‡]Blavatnik School of Computer Science and AI, Tel Aviv University.

[§]Blavatnik School of Computer Science and AI, Tel Aviv University, and Google Research.

[¶]Department of Electrical and Computer Engineering, Technion.

39th Conference on Neural Information Processing Systems (NeurIPS 2025).

Random task orderings are both theoretically motivated and empirically relevant: they closely characterize non-adversarial—and often realistic—task sequences; can be induced algorithmically via random sampling to actively mitigate forgetting; and are implicitly present in standard randomly generated continual learning benchmarks (*e.g.,* split or permuted datasets). These orderings were found to have a remedying effect on forgetting in continual learning, both empirically [27, 20] and theoretically [12, 13, 23, 14]. Under such orderings, the best known dimensionality-independent loss rate for linear regression with jointly realizable tasks is $\mathcal{O}(1/k^{1/4})$ [14], leaving a significant gap from the $\Omega(1/k)$ lower bound that holds for *any* continual learning scheme.

In this work, we analytically reduce both the explicit and implicit regularization schemes to incremental gradient descent, which aligns with SGD under random orderings. We prove that, under jointly realizable tasks, specific choices of fixed and increasing regularization strength schedules yield nearly-optimal and optimal rates of $\mathcal{O}(\log k/k)$ and $\mathcal{O}(1/k)$, respectively.

**Summary of contributions.**    Summarized more technically, our main contributions are:

- We reduce continual linear regression with either *explicit $\ell_2$ regularization* or *finite-step budget* to incremental gradient descent on surrogate losses. These reductions enable unified analysis, even under *arbitrary task orderings* and *non-realizable* settings. Figure 1 visualizes our analytical flow.

- In the realizable case under *random* task orderings, where the best known bound of $\mathcal{O}\left(1/k^{1/4}\right)$ is obtained via an *unregularized* continual scheme:

  - We prove that a carefully set, *fixed* regularization strength yields a *near-optimal* worst-case expected loss of $\mathcal{O}(\log k/k)$.

  - We introduce and analyze a generalized form of SGD for time-varying objectives and use it to propose an *increasing* regularization schedule that achieves the *optimal* rate of $\mathcal{O}(1/k)$, closing the existing gap between upper and lower bounds. See Table 1 for a summary.

Table 1: **Loss rates in realizable continual linear regression** (based on Table 1 of Evron et al. [12]). Upper bounds apply to any $M$ jointly realizable tasks. Lower bounds indicate *worst cases* attained by specific constructions. Bounds for random orderings apply to the *expected* loss. We omit unavoidable scaling terms and constant multiplicative factors (which are mild).

**Notation:** $k =$ iterations; $d =$ dimensions; $\bar{r}, r_{\max} =$ average/maximum data matrix ranks; $a \wedge b \triangleq \min(a, b)$.

| Bound | Regularization | Paper / Ordering | Random with Replacement | Cyclic |
|-------|----------------|------------------|-------------------------|--------|
| Upper | Unregularized | Evron et al. [12] | $\dfrac{d - \bar{r}}{k}$ | $\dfrac{M^2}{\sqrt{k}} \wedge \dfrac{M^2(d - r_{\max})}{k}$ |
| | | Kong et al. [26] | — | $\dfrac{M^3}{k}$ |
| | | Evron et al. [14] | $\dfrac{1}{\sqrt[4]{k}} \wedge \dfrac{\sqrt{d - \bar{r}}}{k} \wedge \dfrac{\sqrt{M\bar{r}}}{k}$ | — |
| | Fixed (explicit) | C&D [5] | — | $\dfrac{M\sqrt{\log(k/M)}}{k}$ |
| | Fixed | **Ours** | $\dfrac{\log k}{k}$ | — |
| | Increasing | **Ours** | $\dfrac{1}{k}$ | — |
| Lower | Unregularized | Evron et al. [12] | $\dfrac{1}{k}$ (∗) | $\dfrac{M^2}{k}$ |
| | Any | **Ours** | $\dfrac{1}{k}$ (∗∗) | — |

(∗) They did not explicitly present such lower bounds, but the 2-task construction from their proof of Theorem 10, can yield a $\Theta(1/k)$ random behavior by cloning those 2 tasks $\lfloor M/2 \rfloor$ times for any general $M$.
(∗∗) While the proof is standard, we are not aware of an explicit statement in the literature.

## 2 Setting: Continual linear regression with explicit or implicit regularization

We focus on the widely studied continual linear regression setting, which, despite its simplicity, often reveals key phenomena and interactions in continual learning [e.g., 11, 12, 31, 16, 36, 28, 48, 17].

**Notation.** Bold symbols are reserved for matrices and vectors. Denote the Euclidean (vectors) or spectral (matrices) norm by $\|\cdot\|$, and the Moore-Penrose inverse by $\mathbf{X}^+$. Finally, denote $[n] = 1, ..., n$.

Throughout the paper, the learner is given access to a *task collection* of $M$ linear regression tasks, that is, $(\mathbf{X}_1, \mathbf{y}_1), \ldots, (\mathbf{X}_M, \mathbf{y}_M)$, where $\mathbf{X}_m \in \mathbb{R}^{n_m \times d}$ and $\mathbf{y}_m \in \mathbb{R}^{n_m}$. We define the data "radius" as $R \triangleq \max_{m \in [M]} \|\mathbf{X}_m\|_2$. Over $k$ iterations, tasks are presented sequentially according to a *task ordering* $\tau \colon [k] \to [M]$. The learner aims to accumulate expertise, quantified by the objective below.

**Definition 2.1** (Average loss). The average—or population—loss is defined as the mean training loss across *all* individual tasks $m \in M$. That is,

$$\mathcal{L}(\mathbf{w}) \triangleq \frac{1}{M} \sum_{m=1}^{M} \mathcal{L}(\mathbf{w}; m) \triangleq \frac{1}{2M} \sum_{m=1}^{M} \|\mathbf{X}_m \mathbf{w} - \mathbf{y}_m\|^2 .$$

*Remark* 2.2 (Forgetting and seen-task loss). Prior work analyzed not only the loss over *all* tasks but also the forgetting, or loss on *seen* tasks. Under the random orderings considered here, all of these quantities are typically close. We thus focus on average loss and discuss the others in Section 4.3.

**Explicit regularization.** A large body of practical continual learning research focuses on mitigating forgetting by *explicitly* penalizing changes in parameter space [e.g., 25, 47, 2, 6]. Many employ regularization terms based on Fisher information [4], though others have found empirically that $\ell_2$ (isotropic) regularization often performs comparably well [32, 40]. Following recent theoretical work [e.g., 28, 13, 5], we also focus on isotropic regularizers but discuss alternatives in Section 4.3.

---
**Scheme 1** Regularized continual linear regression

---
**Input:** Regression tasks $\{(\mathbf{X}_m, \mathbf{y}_m)\}_{m=1}^{M}$, task ordering $\tau$, regularization coefficients $(\lambda_t)_{t=1}^{k}$.

Initialize $\mathbf{w}_0 = \mathbf{0}_d$
For each iteration $t = 1, \ldots, k$:
$\quad \mathbf{w}_t \leftarrow \arg\min_{\mathbf{w}} \left\{ \frac{1}{2} \|\mathbf{X}_{\tau_t} \mathbf{w} - \mathbf{y}_{\tau_t}\|^2 + \frac{\lambda_t}{2} \|\mathbf{w} - \mathbf{w}_{t-1}\|^2 \right\}$
Output $\mathbf{w}_k$

---

*Remark* 2.3 (Unregularized first task). Our analysis is also valid for the common choice $\lambda_1 \to 0$.

While the continual update step above admits a closed-form solution—useful for theoretical analysis [e.g., 28]—our paper does not directly leverage it. Instead, in Section 3, we reduce this step—which solves an *entire* task—to a *single* gradient step, thus enabling last-iterate SGD analysis of the scheme.

**Implicit regularization.** Practically, it is common to minimize the current task's *unregularized* loss with a gradient algorithm for a *finite* number of steps (*e.g.,* in [24]; in contrast to theoretically learning to convergence [12, 14]). This *implicitly* regularizes the model, even in stationary settings [1, 41]. Recently, it has attracted theoretical interest in continual setups [23, 48].

---
**Scheme 2** Continual linear regression with finite step budgets

---
**Input:** Regression tasks $\{(\mathbf{X}_m, \mathbf{y}_m)\}_{m=1}^{M}$, task ordering $\tau$, inner step counts and sizes $(N_t, \gamma_t)_{t=1}^{k}$.

Initialize $\mathbf{w}_0 = \mathbf{0}_d$
For each task $t = 1, \ldots, k$:
$\quad$ Initialize $\mathbf{w}^{(0)} \leftarrow \mathbf{w}_{t-1}$
$\quad$ For $s = 1, \ldots, N_t$:    # Perform $N_t$ gradient steps on the current task's unregularized loss.
$\quad\quad \mathbf{w}^{(s)} \leftarrow \mathbf{w}^{(s-1)} - \gamma_t \nabla \frac{1}{2} \|\mathbf{X}_{\tau_t} \mathbf{w}^{(s-1)} - \mathbf{y}_{\tau_t}\|^2$
$\quad \mathbf{w}_t \leftarrow \mathbf{w}^{(N_t)}$
Output $\mathbf{w}_k$

---

**Regularization strength.** The coefficients $\lambda_t$ and step counts $N_t$ in Schemes 1 and 2 control the "regularization strength" and determine how well the current loss is minimized—often seen as tuning the *stability-plasticity* trade-off [18, 45]. Our paper identifies choices that yield improved bounds.

## 3 Regularized continual linear regression reduces to Incremental GD

Evron et al. [14] proved a reduction from *unregularized* continual linear regression to a "stepwise-optimal" SGD scheme, where a *single* SGD step corresponds to solving an *entire* task. This has allowed them to use last-iterate SGD analysis to study continual learning, as we do in Section 4.

We define the Incremental Gradient Descent (IGD) scheme to cast both Schemes 1 and 2 within a unified framework, enabling a common analysis. The reductions and the flow in which we employ them are illustrated in Figure 1. At each iteration $t$, the algorithm performs a gradient step on the time-varying smooth convex function $f^{(t)}(\cdot; \tau_t)$, selected by the ordering $\tau$, using step size $\eta_t$.

---

**Scheme 3** Incremental Gradient Descent for smooth, convex, time-varying functions

---

**Input:** Smooth, convex, time-varying functions $\left\{ f^{(t)}(\cdot; m) \right\}_{m=1}^{M}$, ordering $\tau$, step sizes $(\eta_t)_{t=1}^{k}$

Initialize $\mathbf{w}_0 \in \mathbb{R}^d$
For each iteration $t = 1, \ldots, k$:
    $\mathbf{w}_t \leftarrow \mathbf{w}_{t-1} - \eta_t \nabla f^{(t)}(\mathbf{w}_{t-1}; \tau_t)$   # Perform a single gradient step on the current objective.
Output $\mathbf{w}_k$

---

We present two reductions that cast regularized and budgeted continual regression as special cases of incremental gradient descent. Proofs for this section are provided in Appendix C.

**Reduction 1** (Regularized Continual Regression ⇒ Incremental GD). *Consider $M$ regression tasks $\{(\mathbf{X}_m, \mathbf{y}_m)\}_{m=1}^{M}$ seen in any ordering $\tau$. Then, each iteration $t \in [k]$ of regularized continual linear regression with a coefficient $\lambda_t > 0$ is equivalent to an IGD step on $f_r^{(t)}(\cdot; \tau_t)$ with step size $\eta_t > 0$, where $f_r^{(t)}(\mathbf{w}; m) \triangleq \frac{1}{2}\left\| \sqrt{\mathbf{A}_m^{(t)}}(\mathbf{w} - \mathbf{X}_m^+ \mathbf{y}_m) \right\|^2$ for some $\mathbf{A}_m^{(t)}$ depending on $\lambda_t, \eta_t$. That is, the iterates of Schemes 1 and 3 coincide.*

**Reduction 2** (Budgeted Continual Regression ⇒ Incremental GD). *Consider $M$ regression tasks $\{(\mathbf{X}_m, \mathbf{y}_m)\}_{m=1}^{M}$ seen in any ordering $\tau$. Then, each iteration $t \in [k]$ of budgeted continual linear regression with $N_t \in \mathbb{N}$ inner steps of size $\gamma_t \in \left(0, 1/R^2\right)$ is equivalent to an IGD step on $f_b^{(t)}(\cdot; \tau_t)$ with step size $\eta_t > 0$, where $f_b^{(t)}(\mathbf{w}; m) \triangleq \frac{1}{2}\left\| \sqrt{\mathbf{A}_m^{(t)}}(\mathbf{w} - \mathbf{X}_m^+ \mathbf{y}_m) \right\|^2$ for some $\mathbf{A}_m^{(t)}$ depending on $N_t, \gamma_t, \eta_t$. That is, the iterates of Schemes 2 and 3 coincide.*

**Proof idea.** The updates $(\mathbf{w}_{t-1} - \mathbf{w}_t)$ in Schemes 1 and 2 are affine in $\mathbf{w}_{t-1}$, and thus correspond to gradients of quadratic functions. By setting $\mathbf{A}_m^{(t)} = \frac{1}{\eta_t}\left(\mathbf{I}_d - \lambda_t \left(\mathbf{X}_m^\top \mathbf{X}_m + \lambda_t \mathbf{I}_d\right)^{-1}\right)$ in Reduction 1, and $\mathbf{A}_m^{(t)} = \frac{1}{\eta_t}\left(\mathbf{I}_d - \left(\mathbf{I}_d - \gamma_t \mathbf{X}_m^\top \mathbf{X}_m\right)^{N_t}\right)$ in Reduction 2, each of those updates coincides with an IGD step on the quadratic surrogate $f^{(t)}(\mathbf{w}; m) = \frac{1}{2}\left\| \sqrt{\mathbf{A}_m^{(t)}}(\mathbf{w} - \mathbf{X}_m^+ \mathbf{y}_m) \right\|^2$.

Next, we establish key properties of the surrogate objectives $f_r^{(t)}, f_b^{(t)}$, which hold *regardless of task ordering or realizability*. Importantly, they enable last-iterate GD analysis for continual regression.

**Lemma 3.1** (Properties of the IGD objectives). *For $t \in [k]$, define $f_r^{(t)}, f_b^{(t)}$ as in Reductions 1 and 2, and recall the data radius $R \triangleq \max_{m \in [M]} \|\mathbf{X}_m\|_2$.*

*(i)* $f_r^{(t)}, f_b^{(t)}$ *are both convex and $\beta$-smooth[1] for $\beta_r^{(t)} \triangleq \frac{1}{\eta_t} \frac{R^2}{R^2 + \lambda_t}$, $\beta_b^{(t)} \triangleq \frac{1}{\eta_t}\left(1 - (1 - \gamma_t R^2)^{N_t}\right)$.*

*(ii) Both functions bound the "excess" loss from both sides, i.e., $\forall \mathbf{w} \in \mathbb{R}^d, \forall t \in [k], \forall m \in [m]$,*

$$\lambda_t \eta_t \cdot f_r^{(t)}(\mathbf{w}; m) \ \leq \ \mathcal{L}(\mathbf{w}; m) - \min_{\mathbf{w}'} \mathcal{L}(\mathbf{w}'; m) \ \leq \ \frac{R^2}{\beta_r^{(t)}} \cdot f_r^{(t)}(\mathbf{w}; m),$$

$$\frac{\eta_t}{\gamma_t N_t} \cdot f_b^{(t)}(\mathbf{w}; m) \ \leq \ \mathcal{L}(\mathbf{w}; m) - \min_{\mathbf{w}'} \mathcal{L}(\mathbf{w}'; m) \ \leq \ \frac{R^2}{\beta_b^{(t)}} \cdot f_b^{(t)}(\mathbf{w}; m).$$

*(iii) Finally, when the tasks are jointly realizable (see Assumption 4.1), the same $\mathbf{w}_\star$ minimizes all surrogate objectives simultaneously. That is,*

$$\mathcal{L}(\mathbf{w}_\star; m) = f_r^{(t)}(\mathbf{w}_\star; m) = f_b^{(t)}(\mathbf{w}_\star; m) = 0, \quad \forall t \in [k], \forall m \in [M].$$

---

[1] A function $h \colon \mathbb{R}^d \to \mathbb{R}$ is $\beta$-smooth when $\|\nabla h(y) - \nabla h(x)\| \leq \beta \|y - x\|$ for all $x, y \in \mathbb{R}^d$.

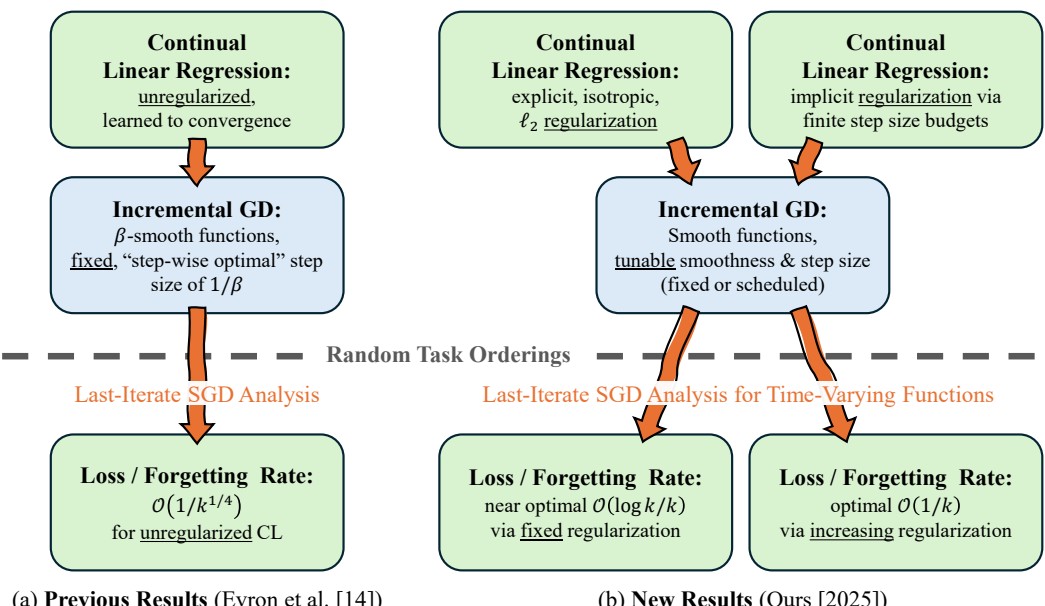

(a) **Previous Results** (Evron et al. [14])    (b) **New Results** (Ours [2025])

Figure 1: **Schematic overview of our contributions compared to prior results.** Evron et al. [14] reduce unregularized continual linear regression to incremental gradient descent on a surrogate objective with fixed smoothness. They then analyze the last iterate of SGD to derive a loss rate of $\mathcal{O}(1/k^{1/4})$ under random task orderings. In contrast, we show that adding explicit or implicit regularization enables tuning the smoothness of the corresponding surrogate objective. Importantly, this added flexibility allows a more nuanced last-iterate analysis: a well-tuned fixed regularization strength yields a near-optimal $\mathcal{O}(\log k/k)$ rate, while a specific increasing schedule achieves the first $\mathcal{O}(1/k)$ rate for continual linear regression under random orderings.

## 4 Rates for realizable continual linear regression in random orderings

**Jointly realizable regression.** In this section, we focus on a setting in which the training data of all tasks can be perfectly fit by a single predictor—a common assumption[2] in theoretical continual learning [e.g., 12, 13, 26, 17, 23, 14]. This assumption simplifies analysis by allowing all iterates to be compared to a fixed predictor, ruling out task collections with inherent contradictions. Such realizability often holds in highly overparameterized deep networks, which can typically be optimized to arbitrarily low training loss. In the neural tangent kernel (NTK) regime [22, 7], such networks exhibit effectively linear dynamics that closely align with our analysis.

**Assumption 4.1** (Joint realizability of training data). There exists an *offline* solution $\mathbf{w}_\star \in \mathbb{R}^d$ s.t.,

$$\mathbf{X}_m \mathbf{w}_\star = \mathbf{y}_m, \quad \forall m \in [M].$$

We note in passing that even under this assumption, models still suffer from forgetting prior expertise—sometimes catastrophically—as thoroughly discussed in Evron et al. [12].

**Random task orderings.** We study random orderings as a natural model of non-adversarial task sequences [38]. Such orderings avoid worst-case pathologies and allow reductions to standard stochastic tools. They are implicitly used when generating common random benchmarks (*e.g.,* permuted or split datasets), and can also be induced algorithmically by random sampling. These settings have been studied empirically [27, 20] and theoretically [12, 13, 23, 14]. Table 1 compares known rates under random and cyclic orderings.

**Definition 4.2** (Random task ordering). A random task ordering samples tasks uniformly from the collection $[M]$. That is, $\tau_1, \ldots, \tau_k \sim \text{Unif}\,([M])$, with or without replacement.

---

[2]Other theoretical works similarly assume an underlying linear model, but allow additive label noise. This, however, almost invariably requires assuming either i.i.d. features [16, 31, 3] or commutative covariance matrices across tasks [28, 29, 48]—whereas we allow *arbitrary* data matrices, enabling worst-case analysis.

**An immediate lower bound.** Under random ordering with replacement, no algorithm can achieve a worst-case expected loss convergence rate faster than $\Omega(1/k)$. This result, which stems from the uncertainty over unseen tasks, is formally established in Theorem B.1 and serves as a baseline for evaluating the tightness of our upper bounds.

Lastly, throughout the section, we use the data radius $R \triangleq \max_{m \in [M]} \|\mathbf{X}_m\|_2$.

## 4.1 Near optimal rates via fixed, horizon-dependent regularization strength

We apply last-iterate convergence results for SGD to the surrogate losses used by IGD under random orderings. Our analysis first considers the fixed-regularization setting, which is common in practice and—unlike our result with increasing regularization—extends cleanly to random orderings *without* replacement (see Remark 4.5). Specifically, using the results of Evron et al. [14] together with the smoothness and upper bound from our Lemma 3.1, we establish the following.

**Lemma 4.3** (Rates for fixed regularization strength). *Assume a random with-replacement ordering over jointly realizable tasks. Then, for each of Schemes 1 and 2, the expected loss after $k \geq 1$ iterations is upper bounded as:*

(i) *Fixed coefficient: For Scheme 1 with a regularization coefficient $\lambda > 0$,*

$$\mathbb{E}_\tau \mathcal{L}\left(\mathbf{w}_k\right) \leq \frac{e\left\|\mathbf{w}_0 - \mathbf{w}_\star\right\|^2 R^2}{2 \cdot \frac{R^2}{R^2+\lambda} \cdot \left(2 - \frac{R^2}{R^2+\lambda}\right) \cdot k^{1 - \frac{R^2}{R^2+\lambda}\left(1 - \frac{R^2}{4(R^2+\lambda)}\right)}} \; ;$$

(ii) *Fixed budget: For Scheme 2 with step size $\gamma \in (0, 1/R^2)$ and budget $N \in \mathbb{N}$,*

$$\mathbb{E}_\tau \mathcal{L}\left(\mathbf{w}_k\right) \leq \frac{e\left\|\mathbf{w}_0 - \mathbf{w}_\star\right\|^2 R^2}{2 \cdot \left(1 - (1 - \gamma R^2)^{2N}\right) \cdot k^{1 - \left(1 - (1 - \gamma R^2)^N\right)\left(1 - \frac{1 - (1 - \gamma R^2)^N}{4}\right)}} \; .$$

All proofs for this subsection are provided in App. D.

The rates established in Lemma 4.3 raise a natural question: *What choice of the regularization strength—i.e., the regularization coefficient $\lambda$ or step count $N$—achieves the tightest bound?*

**Corollary 4.4** (Near-optimal rates via fixed regularization strength). *Assume a random with-replacement ordering over jointly realizable tasks. When the regularization strengths in Lemma 4.3 are set as follows:*

(i) *Fixed coefficient: For Scheme 1, set regularization coefficient $\lambda \triangleq R^2(\ln k - 1)$;*

(ii) *Fixed budget: For Scheme 2, choose step size $\gamma \in (0, 1/R^2)$ and set budget $N \triangleq \frac{\ln\left(1 - \frac{1}{\ln k}\right)}{\ln(1 - \gamma R^2)}$;*

*Then, under either Scheme 1 or Scheme 2, the expected loss after $k \geq 2$ iterations is bounded as:*

$$\mathbb{E}_\tau \mathcal{L}\left(\mathbf{w}_k\right) \leq \frac{5\left\|\mathbf{w}_0 - \mathbf{w}_\star\right\|^2 R^2 \ln k}{k} \; .$$

*Remark* 4.5 (Extension to without replacement orderings). The rates in Lemma 4.3 and Corollary 4.4 extend to random orderings without replacement; see App. D for details.

This marks a significant improvement over the $\mathcal{O}(1/k^{1/4})$ rate established by Evron et al. [14] for the unregularized scheme. By tuning the regularization strength, we gain control over the smoothness of the surrogate losses $f_r^{(t)}$ and $f_b^{(t)}$ in Reductions 1 and 2, allowing us to attain the $\mathcal{O}(\log k/k)$ rate that is optimal within the SGD framework used in their analysis. In contrast, their unregularized scheme lacked this flexibility, which made achieving such rates considerably more difficult and potentially out of reach. A similar rate can also be derived from the last-iterate bounds of Varre et al. [43], as the smoothness induced by our choice of regularization falls within the applicable regime of their results.

While the rate we obtained in the corollary is closer to the lower bound of $\Omega(1/k)$, a gap remains. This leaves an open question: *can regularization be used to match the known lower bound*? In the next section, we develop techniques to answer this question.

## 4.2 Optimal rates via increasing regularization

We present the first result in continual linear regression that achieves the optimal rate for the last iterate, matching the known lower bound. This is obtained by employing a *schedule* in which the regularization strength increases over time. We discuss these findings and their connections to prior work in Section 6. All proofs for this subsection are provided in App. E.

**Theorem 4.6** (Optimal rates for increasing regularization). *Assume a random with-replacement ordering over jointly realizable tasks. Consider either Scheme 1 or Scheme 2 with the following time-dependent schedules:*

(i) *Scheduled coefficient: For Scheme 1, set regularization coefficient* $\lambda_t = \dfrac{13R^2}{3} \cdot \dfrac{k+1}{k-t+2}$;

(ii) *Scheduled budget:*

*For Scheme 2, choose step sizes* $\gamma_t \in (0, 1/R^2)$ *and set budget* $N_t = \dfrac{3}{13\gamma_t R^2} \cdot \dfrac{k-t+2}{k+1}$;

*Then, under either Scheme 1 or Scheme 2, the expected loss after* $k \geq 2$ *iterations is bounded as:*

$$\mathbb{E}_\tau \mathcal{L}(\mathbf{w}_k) \leq \frac{20 \left\| \mathbf{w}_0 - \mathbf{w}_\star \right\|^2 R^2}{k+1}.$$

**Proof technique: Last-iterate analysis for time-varying objectives.** Establishing the theorem requires a novel last-iterate bound in stochastic optimization, as no existing analysis yields a $\mathcal{O}(1/k)$ guarantee for last-iterate convergence in the realizable setting. A standard path to such rates is to use a decreasing step-size schedule. However, our setting is more nuanced: the quantities we control are the regularization strengths—*i.e.*, the regularization coefficient or step budget in Scheme 1 or 2—which inherently modify the surrogate objectives $f_r^{(t)}$ and $f_b^{(t)}$ in Reductions 1 and 2.

To handle this, we analyze SGD applied to *time-varying objectives*—a generalization of standard SGD. For this analysis to yield meaningful guarantees, the evolving surrogates must closely approximate the original loss. Indeed, this condition holds, as verified by Lemma 3.1, thus enabling the application of the next lemma.[3]

**Lemma 4.7** (SGD bound for time-varying distributions). *Assume* $\tau$ *is a random with-replacement ordering over* $M$ *jointly realizable convex and* $\beta$-smooth loss functions $f(\cdot; m) \colon \mathbb{R}^d \to \mathbb{R}$. *Define the average loss* $f(\mathbf{w}) \triangleq \mathbb{E}_{m \sim \tau} f(\mathbf{w}; m)$. *Let* $k \geq 2$, *and suppose* $\{ f^{(t)}(\cdot; m) \mid t \in [k], m \in [M] \}$ *are time-varying surrogate losses that satisfy:*

(i) *Smoothness and convexity:* $f^{(t)}(\cdot; m)$ *are* $\beta$-smooth and convex for all $m \in [M], t \in [k]$;

(ii) *There exists a weight sequence* $\nu_1, \ldots, \nu_k$ *such that for all* $m \in [M], t \in [k], \mathbf{w} \in \mathbb{R}^d$:

$$f^{(t)}(\mathbf{w}; m) - f^{(t)}(\mathbf{w}_\star; m) \leq f(\mathbf{w}; m) - f(\mathbf{w}_\star; m) \leq (1+\nu_t \beta)(f^{(t)}(\mathbf{w}; m) - f^{(t)}(\mathbf{w}_\star; m));$$

(iii) *Joint realizability:*

$$\mathbf{w}_\star \in \cap_{t \in [k]} \cap_{m \in [M]} \arg \min_{\mathbf{w}} f^{(t)}(\mathbf{w}; m); \quad \forall m \in [M], t \in [k], f^{(t)}(\mathbf{w}_\star; m) = f(\mathbf{w}_\star; m).$$

*Then, IGD (Scheme 3) with a diminishing step size that satisfies* $\nu_t \leq \eta_t = \eta \left( \frac{k-t+2}{k+1} \right)$, $\forall t \in [k]$ *for some* $\eta \leq 3/(13\beta)$, *guarantees the following expected loss bound:*

$$\mathbb{E} f(\mathbf{w}_k) - f(\mathbf{w}_\star) \leq \frac{9}{2\eta(k+1)} \left\| \mathbf{w}_0 - \mathbf{w}_\star \right\|^2 .$$

*In particular, for* $\eta = \frac{3}{13\beta}$ *we obtain,*

$$\mathbb{E} f(\mathbf{w}_k) - f(\mathbf{w}_\star) \leq \frac{20\beta \left\| \mathbf{w}_0 - \mathbf{w}_\star \right\|^2}{k+1} .$$

---

[3]While Lemma 4.7 serves Theorem 4.6 in our least-squares setting, it is stated more generally to apply to a broader family of objectives.

### 4.3 *Do not forget forgetting:* **Extension to seen-task loss**

We now take the opportunity to briefly revisit our results through the lens of other quantities of interest beyond the average (training) loss defined in Definition 2.1.

Continual (or lifelong) learning aims to develop systems that accumulate expertise over time—learning from new experiences without *forgetting* previous ones [33, 15]. While mitigating forgetting has long been a central goal in continual learning, practitioners often monitor it indirectly using "positive" metrics, such as average accuracy or performance [37, 25, 30].

In theoretical work, however, it is essential to define such quantities explicitly. Doan et al. [11] defined forgetting at time $k$ as the drift in model *outputs*, *e.g.*, $\frac{1}{k} \sum_{t=1}^{k} \|\mathbf{X}_{\tau_t}(\mathbf{w}_k - \mathbf{w}_t)\|^2$. Nevertheless, this can be large even if the model *improves* between times $t$ and $k$—that is, in the presence of positive *backward transfer*.

An alternative forgetting definition, used, *e.g.,* by Evron et al. [12, 14], Lin et al. [31], is *loss degradation*: $\frac{1}{k} \sum_{t=1}^{k} \mathcal{L}(\mathbf{w}_k; \tau_t) - \mathcal{L}(\mathbf{w}_t; \tau_t) = \frac{1}{2k} \sum_{t=1}^{k} \|\mathbf{X}_{\tau_t}\mathbf{w}_k - \mathbf{y}_{\tau_t}\|^2 - \|\mathbf{X}_{\tau_t}\mathbf{w}_t - \mathbf{y}_{\tau_t}\|^2$. Commonly, such works [12, 17] assume joint realizability (as we do), and also that the model is trained *to convergence* at each step, achieving zero loss on the current task. In that case, forgetting reduces to: $\frac{1}{2k} \sum_{t=1}^{k} \|\mathbf{X}_{\tau_t}\mathbf{w}_k - \mathbf{y}_{\tau_t}\|^2$, which is always non-negative and can be meaningfully upper bounded.

However, in schemes like our regularized approaches (Schemes 1 and 2), where convergence is not achieved despite realizability, loss degradation can be *negative* due to backward transfer. As a result, it is sensitive to worst-case analytical "manipulations" and difficult to analyze theoretically.

We introduce a more suitable alternative: the *seen-task loss*, which quantifies performance on previously encountered tasks. Importantly, this quantity is always non-negative and decreases in the presence of desirable backward transfer.

**Definition 4.8** (Seen-task loss)**.** Let $\tau : [k] \to [M]$ be the task ordering, and let $\mathbf{w}_k$ be the iterate (parameters) after $k$ steps. The *seen-task loss* at step $k$ is defined as

$$\mathcal{L}_{1:k}(\mathbf{w}_k) \triangleq \frac{1}{k} \sum_{t=1}^{k} \mathcal{L}(\mathbf{w}_k; \tau_t).$$

In App. E, we extend Theorem 4.6 from the *average* loss to the *seen-task* loss. Specifically, we show that increasing regularization also achieves an $\mathcal{O}(1/k)$ rate for the expected seen-task loss. But, *is this the optimal rate for seen-task loss?*

The next lemma shows that, at least under explicit isotropic regularization (Scheme 1), it *is* optimal. Proof in App. B. More precisely, under random task orderings, no regularization schedule yields a rate faster than $\mathcal{O}(1/k)$ for the expected seen-task loss. In Section 6, we discuss how non-isotropic regularization—at the cost of additional space complexity—can ensure a seen-task loss of zero.

**Lemma 4.9** (Lower bound for seen-task loss of the isotropic Scheme 1 under any coefficient sequence)**.** *For any $d \geq 2$, initialization $\mathbf{w}_0 \in \mathbb{R}^d$, and regularization coefficient sequence $\lambda_1, \ldots, \lambda_k \geq 0$, there exists a set of jointly realizable linear regression tasks $\{(\mathbf{X}_m, \mathbf{y}_m)\}_{m=1}^{M}$ such that, under a with-replacement random task ordering, Scheme 1 incurs seen-task loss $\mathcal{L}_{1:k}(\mathbf{w}_k) = \Omega(1/k)$ with probability at least $1/10$.*

## 5   Related work

**Explicit regularization.**   Recent theoretical work on continual learning has studied the explicitly regularized scheme (Scheme 1) in continual linear regression settings [28, 48, 29], with several key differences from our work. Like we do, these papers focused on settings where labels stem from an underlying linear model. However, they analyzed the *generalization* loss given *noisy* data, while we analyze the *training* loss given *noiseless* data. Theirs may sound like a "stronger", more permissive setup, but comes at the price of a very restrictive assumption: the expected task covariances $\mathbb{E}\mathbf{X}_1^\top \mathbf{X}_1, \ldots, \mathbb{E}\mathbf{X}_M^\top \mathbf{X}_M$ are assumed to commute. This commutativity removes forgetting due to misaligned feature subspaces across tasks, leaving noise as the sole culprit behind any degradation.

To minimize the expected risk under this assumption, Zhao et al. [48] proposed a regularization weight matrix proportional to the sum of observed task covariances, which—like our proposed

schedule—increases over time. *However, their approach is conceptually distinct from ours:* the mechanism driving their schedule exploits the commutativity assumption, which eliminates task misalignment, whereas our schedule explicitly mitigates degradation caused by such misalignment. As a result, the motivations—and guarantees—behind the two schedules are fundamentally different.

Li et al. [28, 29] focused exclusively on sequences of $M = 2$ tasks. Li et al. [28] derived risk bounds for isotropic regularization (Scheme 1) and highlight a trade-off between forgetting and intransigence. Li et al. [29] demonstrated that, under additional restrictions on the data matrices, there is a trade-off between increased memory usage and the performance of regularized continual linear regression. In all these works, performance degradation is attributed solely to label noise. In contrast, we analyzed interference that arises even in the absence of noise. Accordingly, their focus lies in a complementary regime that does not capture the challenges we address.

**Finite step budgets.**    Two main theoretical works studied the finite budget setting (Scheme 2). Jung et al. [23] analyzed continual linear *classification* under cyclic and random orderings. For cyclic orderings, they provided convergence rate for the loss; and, for random orderings, they only proved asymptotic convergence. Moreover, classification settings can yield different results and conclusions compared to regression settings [see 13]. Zhao et al. [48] analyzed both regularized and budgeted continual linear regression schemes under restrictive assumptions, showing that a carefully constructed, task-dependent regularization matrix can force the iterates of the regularized scheme to match those of the budgeted one. This alignment, however, requires precise knowledge of task covariances and breaks under standard isotropic $\ell_2$ regularization. In contrast, our unified reduction of both schemes to IGD (Section 3) avoids this limitation entirely.

**Proximal method.**    Cai and Diakonikolas [5] analyzed the Incremental Proximal Method (IPM), corresponding to isotropic $\ell_2$ regularization, under *cyclic* orderings. They provided convergence rates for convex smooth or convex Lipschitz losses with bounded noise, but their guarantees only become meaningful after multiple full sweeps (or epochs) over the task sequence. In contrast, we analyze the *random* orderings and establish nontrivial—and even *optimal*—guarantees without requiring repeated passes. See Section 6 for a comparison with our regularization schedules.

## 6    Discussion

Our work reduces regularized continual linear regression—whether using explicit or implicit regularization—to incremental gradient descent on smooth surrogate losses. This reduction enables last-iterate SGD analysis under random task orderings and yields significantly improved convergence rates. Specifically, we show that a suitable fixed regularization strength achieves a near-optimal rate of $\mathcal{O}(\log k/k)$, while a carefully chosen increasing strength schedule achieves, for the first time, an $\mathcal{O}(1/k)$ rate, matching the known lower bound. See Figure 1 and Table 1 for a summary of the reductions and improved rates.

**Regularization strength scheduling.**    In Section 4.2, we derived an optimal regularization schedule in which the regularization strength increases with each task. This implies that the parameters change progressively less over time. Interestingly, such an attenuation in "synaptic plasticity" is also observed in biological systems: the rate at which synapses grow or shrink in response to sensory stimulation [44] or motor learning [21] significantly decreases over time as the brain matures [35].

In continual learning, many papers practically set a fixed regularization coefficient $\lambda$ through simple hyperparameter tuning. However, non-isotropic weighting schemes often encode an implicit scale in the weighting matrices they compute. Methods such as EWC [25] and Path Integral [47] are particularly sensitive to $\lambda$, as their weighting matrices tend to have low magnitude early in training and may increase over time [see 10]. This initially low regularization strength was considered problematic by some [e.g., 6] and was even canceled algorithmically, as it allows excessive plasticity in early tasks. Yet, one may argue that high plasticity is desirable in the beginning of *long* task sequences, where substantial expertise remains to be acquired. Our analysis in Theorem 4.6 supports this intuition, showing that in such cases, an increasing regularization schedule yields optimal upper bounds under random task orderings. See also the findings and discussion in Mirzadeh et al. [34] on the effects of a decaying step size, which—as noted in our Section 2—corresponds to an increasing regularization strength.

Analytically, Evron et al. [13] showed that in continual linear models for binary classification, increasing the regularization coefficient can be *harmful* to convergence guarantees (see their Example 3). However, their analysis applies only to *weakly* regularized schemes (where $\lambda_t \to 0$ for all $t$), and the problematic schedule they presented increases the coefficient at a doubly-exponential rate—in contrast to our Theorem 4.6 which utilizes finite, and relatively large, coefficients that increase *linearly*. Under *cyclic* orderings over linear regression tasks, solved with explicit regularization (Scheme 1), the analysis of Cai and Diakonikolas [5] dictates a *fixed* coefficient $\lambda = 2MR^2\sqrt{\ln(k/M)}$. In contrast, under *random* orderings, our *fixed* variant in Section 4.1 sets $\lambda = R^2(\ln k - 1)$. While both choices grow at most logarithmically with $k$, theirs grows with the number of tasks $M$, making it less suitable for "single-epoch" settings—though effective in the multi-epoch regime that they studied.

To complement our analysis, App. A empirically examines a simple two-task setup trained under the explicit-regularization scheme with random task ordering. Interestingly, in this non-worst-case setting, the optimal fixed regularization, obtained by minimizing the expected loss after $k$ steps, exhibits an approximately *linear* dependence on $k$, in contrast to the *logarithmic* scaling we predicted for the *worst case*.

**Non-isotropic explicit regularization.**   Throughout the paper, we assumed Scheme 1 uses isotropic $\ell_2$ regularization. Such regularization often performs competitively with weighted schemes in practice [32, 40]. The latter, widely used in the literature, typically rely on weight matrices derived from Fisher information, often approximated by their diagonal [25, 47, 2, 4]. Theoretically, using the full Fisher matrix from previous tasks requires $\mathcal{O}(d^2)$ memory in the worst case, but guarantees *zero* seen-task loss (Definition 4.8)—that is, complete retention of *past* expertise (see Proposition 5.5 of Evron et al. [13] and Proposition 5 of Peng et al. [36]). Closed-form alternatives include Recursive Least Squares (Chapter 12.2 in 8) and its block variant BRMP [49], which likewise maintain an explicit second-order memory to prevent forgetting. More generally, the rank of the regularization weight matrix can offer a trade-off between memory and forgetting [29].

**Last-iterate convergence of SGD in the realizable smooth setting.**   Our Lemma 4.7, originally proved to leverage the reductions from continual learning to the incremental gradient descent method, also establishes a last-iterate convergence guarantee for a variant of SGD that may be of independent interest. By setting the surrogate functions equal to the original functions, this result yields an $\mathcal{O}(1/k)$ convergence guarantee for convex smooth optimization in the realizable regime, using a linear decay schedule [9]. To our knowledge, this is the first fast-rate guarantee for the last-iterate convergence of SGD in the realizable setting. It not only generalizes prior results specific to least-squares problems [43], but also improves the convergence rate from $O(\log T/T)$ to the optimal $O(1/T)$.

**Future work.**   While we establish *optimal* rates for realizable continual linear regression with regularization under random task orderings, several directions remain open. First, empirical validation on standard continual learning benchmarks would test the practical impact of our regularization strength schedules. Second, extending our reduction-based analysis to simple nonlinear models could reveal whether similar schedules yield optimal convergence in more expressive settings.

Perhaps more fundamentally, it remains an open question whether comparable bounds can be achieved *without* regularization—that is, whether the gap between the upper bound of Evron et al. [14] and the lower bound—both shown in Table 1—can be closed. Clarifying this would shed light on whether regularization is essential for achieving optimal rates, or if it simply serves a role as a convenient analytical device. Addressing these questions may help close the gap between theoretical guarantees and practical continual learning systems. Finally, fully understanding the role of regularization in typical (non-worst-case) settings remains an open question for future work.

## Acknowledgments and Disclosure of Funding

We thank Edward Moroshko (University of Edinburgh) for insightful discussions.

The research of TK has received funding from the European Research Council (ERC) under the European Union's Horizon 2020 research and innovation program (grant agreement No. 101078075). Views and opinions expressed are however those of the author(s) only and do not necessarily reflect those of the European Union or the European Research Council. Neither the European Union nor the

granting authority can be held responsible for them. This work received additional support from the Israel Science Foundation (ISF, grant number 3174/23), from the Israeli Council for Higher Education, and a grant from the Tel Aviv University Center for AI and Data Science (TAD).

The research of DS was funded by the European Union (ERC, A-B-C-Deep, 101039436). Views and opinions expressed are however those of the author only and do not necessarily reflect those of the European Union or the European Research Council Executive Agency (ERCEA). Neither the European Union nor the granting authority can be held responsible for them. DS also acknowledges the support of the Schmidt Career Advancement Chair in AI.

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

## A Empirical Illustration

In this section, we illustrate the dependence of the optimal *fixed* regularization coefficient $\lambda$ on the horizon $k$ and task angle $\theta$ in a simple two-task setup. Two single sample tasks are defined as $\mathbf{x}_1 = (1, 0)$ and $\mathbf{x}_2 = (\cos\theta, \sin\theta)$, trained under the explicit–regularization scheme (Scheme 1) with a random with–replacement ordering (Definition 4.2). For each $(k, \theta)$ we numerically determine the fixed coefficient $\lambda^\star(k; \theta)$ that minimizes the expected loss after $k$ steps. As shown in Figure 2, the optimal regularization grows roughly linearly with $k$, and its magnitude increases with $\theta$–indicating that longer horizons and more misaligned tasks benefit from stronger regularization.

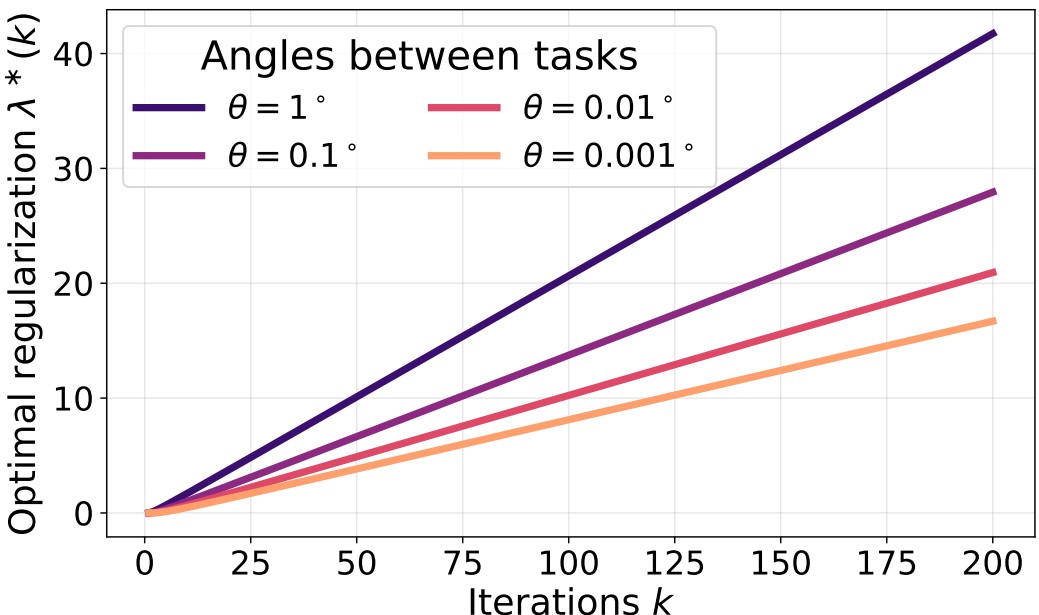

Figure 2: **Optimal fixed regularization grows with horizon and task angle.** Each curve shows $\lambda^\star(k; \theta)$ obtained by minimizing the expected loss after $k$ steps of the explicit-regularization scheme with a constant $\lambda$. We observe an approximately linear growth in $k$ and higher optimal regularization for larger $\theta$.

**Discussion.** The qualitative behavior in Figure 2 aligns with the general principles established in the paper, but the quantitative trend differs. In Section 4, we prove an optimal *time–varying* regularization schedule and a near-optimal *fixed* coefficient for the worst-case analysis over arbitrary task distributions. Here, by contrast, we examine a simple two-task setting, where the empirically optimal fixed regularization $\lambda^\star(k; \theta)$ grows approximately linearly with $k$, rather than logarithmically as predicted by Corollary 4.4. A plausible explanation is that the worst-case distribution inducing the $\log(k)$ scaling is more complex than the two-task setup considered here, so optimizing for this simple distribution yields an inherently different solution from the true worst-case behavior across all distributions.

**Computation.** We compute the expected loss in closed form using an operator formulation similar to that of Evron et al. [14], originally developed to establish their fast dimensionality-dependent rate. At each step of Scheme 1,

$$\mathbf{w}_t = (\mathbf{X}_{\tau_t}^\top \mathbf{X}_{\tau_t} + \lambda\mathbf{I})^{-1}(\mathbf{X}_{\tau_t}^\top \mathbf{y}_{\tau_t} + \lambda\mathbf{w}_{t-1}), \quad \lambda > 0.$$

Under realizability ($\mathbf{y}_m = \mathbf{X}_m \mathbf{w}_\star$), defining $\mathbf{z}_t = \mathbf{w}_t - \mathbf{w}_\star$ gives

$$
\begin{aligned}
\mathbf{z}_t &= (\mathbf{X}_{\tau_t}^\top \mathbf{X}_{\tau_t} + \lambda \mathbf{I})^{-1} (\mathbf{X}_{\tau_t}^\top \mathbf{y}_{\tau_t} + \lambda \mathbf{w}_{t-1}) - \mathbf{w}_\star \\
&= (\mathbf{X}_{\tau_t}^\top \mathbf{X}_{\tau_t} + \lambda \mathbf{I})^{-1} (\mathbf{X}_{\tau_t}^\top \mathbf{X}_{\tau_t} \mathbf{w}_\star + \lambda \mathbf{w}_{t-1}) - \mathbf{w}_\star \\
&= (\mathbf{X}_{\tau_t}^\top \mathbf{X}_{\tau_t} + \lambda \mathbf{I})^{-1} \big((\mathbf{X}_{\tau_t}^\top \mathbf{X}_{\tau_t} + \lambda \mathbf{I})\mathbf{w}_\star + \lambda(\mathbf{w}_{t-1} - \mathbf{w}_\star)\big) - \mathbf{w}_\star \\
&= \mathbf{w}_\star + \lambda (\mathbf{X}_{\tau_t}^\top \mathbf{X}_{\tau_t} + \lambda \mathbf{I})^{-1} (\mathbf{w}_{t-1} - \mathbf{w}_\star) - \mathbf{w}_\star \\
&= \lambda (\mathbf{X}_{\tau_t}^\top \mathbf{X}_{\tau_t} + \lambda \mathbf{I})^{-1} \mathbf{z}_{t-1}.
\end{aligned}
$$

Since $\tau_t$ is sampled independently at each step, taking expectations over the randomness of $\tau$ gives

$$
\mathbb{E}[\mathbf{z}_t \mathbf{z}_t^\top] = \mathbb{E}\big[\lambda (\mathbf{X}_{\tau_t}^\top \mathbf{X}_{\tau_t} + \lambda \mathbf{I})^{-1} \, \mathbb{E}[\mathbf{z}_{t-1} \mathbf{z}_{t-1}^\top] \, \lambda (\mathbf{X}_{\tau_t}^\top \mathbf{X}_{\tau_t} + \lambda \mathbf{I})^{-1}\big].
$$

Starting from $\mathbf{w}_0 = \mathbf{0}$, the initial error is $\mathbf{z}_0 = -\mathbf{w}_\star$, so $\mathbb{E}[\mathbf{z}_0 \mathbf{z}_0^\top] = \mathbf{w}_\star \mathbf{w}_\star^\top$. Define the linear operator

$$
\mathcal{Q}_\lambda : \mathbb{R}^{d \times d} \to \mathbb{R}^{d \times d}, \qquad \mathcal{Q}_\lambda[\mathbf{A}] = \mathbb{E}_\tau \big[\lambda (\mathbf{X}_\tau^\top \mathbf{X}_\tau + \lambda \mathbf{I})^{-1} \, \mathbf{A} \, \lambda (\mathbf{X}_\tau^\top \mathbf{X}_\tau + \lambda \mathbf{I})^{-1}\big].
$$

By independence of $\tau_t$ and $\mathbf{z}_{t-1}$,

$$
\mathbb{E}[\mathbf{z}_t \mathbf{z}_t^\top] = \mathcal{Q}_\lambda \big[\mathbb{E}[\mathbf{z}_{t-1} \mathbf{z}_{t-1}^\top]\big], \qquad \mathbb{E}[\mathbf{z}_k \mathbf{z}_k^\top] = \mathcal{Q}_\lambda^k \big[\mathbf{w}_\star \mathbf{w}_\star^\top\big].
$$

By Definition 2.1, the population loss after $k$ steps is the expected training loss across tasks:

$$
\mathbb{E}[\mathcal{L}(\mathbf{w}_k)] = \frac{1}{2} \mathbb{E}\big[\|\mathbf{X}_\tau \mathbf{w}_k - \mathbf{y}_\tau\|^2\big].
$$

Under realizability ($\mathbf{y}_\tau = \mathbf{X}_\tau \mathbf{w}_\star$), this equals

$$
\mathbb{E}[\mathcal{L}(\mathbf{w}_k)] = \frac{1}{2} \mathbb{E}\big[\|\mathbf{X}_\tau (\mathbf{w}_k - \mathbf{w}_\star)\|^2\big] = \frac{1}{2} \mathbb{E}\big[\mathbf{z}_k^\top \mathbf{X}_\tau^\top \mathbf{X}_\tau \mathbf{z}_k\big].
$$

Taking the expectation inside the trace gives

$$
\mathbb{E}[\mathcal{L}(\mathbf{w}_k)] = \tfrac{1}{2} \text{Tr}\Big(\mathbb{E}[\mathbf{X}_\tau^\top \mathbf{X}_\tau] \, \mathbb{E}[\mathbf{z}_k \mathbf{z}_k^\top]\Big) = \tfrac{1}{2} \text{Tr}\Big(\mathbb{E}[\mathbf{X}_\tau^\top \mathbf{X}_\tau] \, \mathcal{Q}_\lambda^k[\mathbf{w}_\star \mathbf{w}_\star^\top]\Big).
$$

To move $\mathcal{Q}_\lambda^k$ from $\mathbf{w}_\star \mathbf{w}_\star^\top$ onto $\mathbb{E}[\mathbf{X}_\tau^\top \mathbf{X}_\tau]$, expand $\mathcal{Q}_\lambda^k$ along an i.i.d. sequence $\tau_1, \ldots, \tau_k$ and apply linearity of expectation and the cyclic property of the trace:

$$
\begin{aligned}
\text{Tr}\Big(\mathbb{E}[\mathbf{X}_\tau^\top \mathbf{X}_\tau] \, \mathcal{Q}_\lambda^k[\mathbf{w}_\star \mathbf{w}_\star^\top]\Big) &= \mathbb{E}_{\tau_1, \ldots, \tau_k}\Big[\text{Tr}\big(\mathbb{E}[\mathbf{X}_\tau^\top \mathbf{X}_\tau] \, \boldsymbol{\Lambda}_{\tau_k} \cdots \boldsymbol{\Lambda}_{\tau_1} \, \mathbf{w}_\star \mathbf{w}_\star^\top \, \boldsymbol{\Lambda}_{\tau_1} \cdots \boldsymbol{\Lambda}_{\tau_k}\big)\Big] \\
&= \mathbb{E}_{\tau_1, \ldots, \tau_k}\Big[\text{Tr}\big(\boldsymbol{\Lambda}_{\tau_1} \cdots \boldsymbol{\Lambda}_{\tau_k} \, \mathbb{E}[\mathbf{X}_\tau^\top \mathbf{X}_\tau] \, \boldsymbol{\Lambda}_{\tau_k} \cdots \boldsymbol{\Lambda}_{\tau_1} \, \mathbf{w}_\star \mathbf{w}_\star^\top\big)\Big] \\
&= \text{Tr}\Big(\mathcal{Q}_\lambda^k\big[\mathbb{E}[\mathbf{X}_\tau^\top \mathbf{X}_\tau]\big] \, \mathbf{w}_\star \mathbf{w}_\star^\top\Big),
\end{aligned}
$$

where $\boldsymbol{\Lambda}_\tau := \lambda (\mathbf{X}_\tau^\top \mathbf{X}_\tau + \lambda \mathbf{I})^{-1}$. Hence,

$$
\mathbb{E}[\mathcal{L}(\mathbf{w}_k)] = \tfrac{1}{2} \text{Tr}\Big(\mathcal{Q}_\lambda^k\big[\mathbb{E}[\mathbf{X}_\tau^\top \mathbf{X}_\tau]\big] \, \mathbf{w}_\star \mathbf{w}_\star^\top\Big) = \tfrac{1}{2} \mathbf{w}_\star^\top \mathcal{Q}_\lambda^k\big[\mathbb{E}[\mathbf{X}_\tau^\top \mathbf{X}_\tau]\big] \mathbf{w}_\star.
$$

Taking the maximum over all unit teachers $\|\mathbf{w}_\star\| = 1$ gives the worst–case expected loss

$$
L_k^{\text{worst}} = \tfrac{1}{2} \, \lambda_{\max}\Big(\mathcal{Q}_\lambda^k\big[\mathbb{E}[\mathbf{X}_\tau^\top \mathbf{X}_\tau]\big]\Big).
$$

This formulation allows us to compute the expected worst–case loss for Scheme 1 entirely in closed form. All stochastic effects of task ordering are captured analytically through expectations, eliminating the need for Monte-Carlo sampling or repeated randomized experiments.

# B   Proofs of lower bounds

**Theorem B.1.** *Let $d \geq 2$ and $k \geq 2$. Then for any algorithm $\mathcal{A}$ which receives $k$ functions $f_1, f_2, \ldots, f_k : \mathbb{R}^d \to \mathbb{R}$ and outputs a point in $\mathbb{R}^d$, there exists a point $\mathbf{w}_\star \in \mathbb{R}^d$ such that $\|\mathbf{w}_\star\| \leq 1$ and a set of $k$ 1-smooth convex quadratic functions which are minimized at $\mathbf{w}_\star$, $h_1, \ldots, h_k : \mathbb{R}^d \to \mathbb{R}$ such that*

$$\mathbb{E}_{\tau(1), \ldots, \tau(k) \sim \text{Unif}([k]), \mathcal{A}}[F(\mathcal{A}(h_{\tau(1)}, \ldots, h_{\tau(k)})) - F(\mathbf{w}_\star)] = \Omega(1/k),$$

*where $F(\mathbf{w}) \triangleq \mathbb{E}_{i \sim \text{Unif}([k])} h_i(\mathbf{w})$.*

*Proof.* In the following proof we denote with $\mathbf{w}[i]$ the $i$'th coordinate of a vector $\mathbf{w}$. Given an algorithm $\mathcal{A}$, let $h_1(\mathbf{w}) = \frac{1}{2}\mathbf{w}[1]^2$, $h_i = h_1$ for $i = 2, \ldots, k-1$, and $E_B = \{\forall i \in [k] : \tau(i) \neq k\}$ be the bad event where last index is not sampled. Note that as $1 - x \geq 4^{-x}$ for all $x \in [0, \frac{1}{2}]$,

$$\Pr(E_B) = \left(1 - \frac{1}{k}\right)^k \geq \frac{1}{4}.$$

Let $\tilde{\mathbf{w}}$ be the (stochastic) output of $\mathcal{A}(h_1, h_1, \ldots, h_1)$ (when $\mathcal{A}$ is presented with $k$ copies of $h_1$), and let

$$a = \begin{cases} 1 & \text{if } \Pr(\tilde{\mathbf{w}}[2] \leq 0) \geq \frac{1}{2}; \\ -1 & \text{if } \Pr(\tilde{\mathbf{w}}[2] \leq 0) < \frac{1}{2}. \end{cases}$$

Let $h_k(\mathbf{w}) = \frac{1}{2}(\mathbf{w}[2] - a)^2$. Note that all functions are 1-smooth, convex, quadratic, and minimized at $\mathbf{w}_\star = (0, a, 0, \ldots, 0)$, where $\|\mathbf{w}_\star\| \leq 1$. Hence, as $\mathbf{w}_\star$ is a minimizer of $F(\mathbf{w})$,

$$\mathbb{E}[F(\mathcal{A}(h_{\tau(1)}, \ldots, h_{\tau(k)})) - F(\mathbf{w}_\star)] \geq \Pr(E_B)\mathbb{E}[F(\mathcal{A}(h_{\tau(1)}, \ldots, h_{\tau(k)})) - F(\mathbf{w}_\star) \mid E_B]$$

$$= \Pr(E_B)\mathbb{E}[F(\mathcal{A}(h_1, h_1, \ldots, h_1)) - F(\mathbf{w}_\star) \mid E_B]$$
$$\text{(Conditioned on } E_B, h_\tau(i) = h(1) \text{ for all } i\text{)}$$

$$= \Pr(E_B)\mathbb{E}[F(\mathcal{A}(h_1, h_1, \ldots, h_1)) \mid E_B] \qquad\qquad (h_i(\mathbf{w}_\star) = 0 \text{ for all } i)$$

$$\geq \frac{1}{k}\Pr(E_B)\mathbb{E}[h_k(\mathcal{A}(h_1, h_1, \ldots, h_1)) \mid E_B]. \qquad (F(\mathbf{w}) \geq \frac{1}{k}h_i(\mathbf{w}) \text{ for any } i, \mathbf{w})$$

Conditioned on $E_B$, with probability at least $\frac{1}{2}$, $\mathbf{w} = \mathcal{A}(h_1, h_1, \ldots, h_1)$ satisfies $(\mathbf{w}[2] - a)^2 \geq 1$. Thus,

$$\mathbb{E}[F(\mathcal{A}(h_{\tau(1)}, \ldots, h_{\tau(k)})) - F(\mathbf{w}_\star)] \geq \frac{\Pr(E_B)}{4k} \geq \frac{1}{16k} = \Omega(1/k).$$

$\square$

Our next lemma makes use of the Sherman-Morison formula, stated below for completeness.

**Lemma B.2** (Sherman-Morison). *Suppose $\mathbf{X} \in \mathbb{R}^{d \times d}$ is invertible, and $\mathbf{u}, \mathbf{v} \in \mathbb{R}^d$. Then $\mathbf{X} + \mathbf{u}\mathbf{v}^\top$ is invertible iff $1 + \mathbf{v}^\top \mathbf{X}^{-1}\mathbf{u} \neq 0$, in which case is holds that:*

$$\left(\mathbf{X} + \mathbf{u}\mathbf{v}^\top\right)^{-1} = \mathbf{X}^{-1} - \frac{\mathbf{X}^{-1}\mathbf{u}\mathbf{v}^\top\mathbf{X}^{-1}}{1 + \mathbf{v}^\top\mathbf{X}^{-1}\mathbf{u}}.$$

**Recall Lemma 4.9 — lower bound for seen-task loss under Scheme 1.** For any $d \geq 2$, initialization $\mathbf{w}_0 \in \mathbb{R}^d$, and regularization coefficient sequence $\lambda_1, \ldots, \lambda_k \geq 0$, there exists a set of jointly realizable linear regression tasks $\{(\mathbf{X}_m, \mathbf{y}_m)\}_{m=1}^M$ such that, under a with-replacement random task ordering, Scheme 1 incurs seen-task loss $\mathcal{L}_{1:k}(\mathbf{w}_k) = \Omega(1/k)$ with probability at least $1/10$.

*Proof.* Let $k \geq 9$, and let $\lambda_1, \ldots, \lambda_k \geq 0$ be any regularization sequence. For simplicity, we set $M = k$, but the proof can be easily extended to $M > k$. Let $f_1(\mathbf{w}) = \frac{1}{2}(\mathbf{e}_2^\top \mathbf{w})^2$, where $\mathbf{e}_2 = (0, 1, 0, \ldots, 0)^\top$, and $f_2(\mathbf{w}) = \frac{1}{2}(\mathbf{x}^\top \mathbf{w})^2$, where $\mathbf{x} = (\sqrt{1-\alpha^2}, \alpha, 0, \ldots, 0)^\top$ for some $\alpha \in [0, 1]$. Note that these can be represented as tasks $\{(\mathbf{e}_2, 0), (\mathbf{x}, 0)\}$ with $R = 1$.

Consider the uniform distribution over the set $\{f_1, \ldots, f_1, f_2\}$ of size $k$, such that $f_1$ is sampled with probability $1 - \frac{1}{k}$ and $f_2$ is sampled with probability $\frac{1}{k}$. Let $E_B$ be the "bad" event where $f_2$ is sampled exactly once, and note that using the inequality $1 - x \geq 4^{-x}$ which holds for all $x \in [0, \frac{1}{2}]$,

$$\Pr(E_B) = k \cdot \frac{1}{k} \cdot \left(1 - \frac{1}{k}\right)^{k-1} = \left(1 - \frac{1}{k}\right)^k \frac{k}{k-1} \geq \frac{1}{4}.$$

The rest of the analysis will be conditioned on the "bad" event. Let $\lambda > 0$, and note that for any $\mathbf{w}$,

$$\arg\min_{\mathbf{w}'}\{f_1(\mathbf{w}') + \tfrac{\lambda}{2}\|\mathbf{w}' - \mathbf{w}\|^2\} = \left(\mathbf{e}_2\mathbf{e}_2^\top + \lambda\mathbf{I}\right)^{-1}(\lambda\mathbf{w}) = \mathbf{w} - \frac{(\mathbf{w}^\top \mathbf{e}_2)}{\lambda+1}\mathbf{e}_2,$$

where the second equality follows from Lemma B.2. The case of $\lambda = 0$ is treated as the update above with $\lambda = 0$, and similarly,

$$\arg\min_{\mathbf{w}'}\{f_2(\mathbf{w}') + \tfrac{\lambda}{2}\|\mathbf{w}' - \mathbf{w}\|^2\} = \mathbf{w} - \frac{(\mathbf{w}^\top \mathbf{x})}{\lambda+1}\mathbf{x}.$$

Starting at $\mathbf{w}_0 = (1, 0)^\top$, the iterates will not move until encountered with $f_2$. Denote with $t_0$ this step. Thus,

$$\mathbf{w}_{t_0} = \left(1 - \frac{1-\alpha^2}{\lambda_{t_0}+1}, -\frac{\alpha\sqrt{1-\alpha^2}}{\lambda_{t_0}+1}\right)^\top.$$

From now on, we only observe $f_1$, so the first coordinate of $\mathbf{w}_k$ for $k > t_0$, which we denote as $\mathbf{w}_t[1]$, is

$$\mathbf{w}_k[1] = \mathbf{w}_{k-1}[1] - \frac{(\mathbf{w}_{k-1}^\top \mathbf{e}_2)}{\lambda+1}\mathbf{e}_2[1] = \mathbf{w}_{k-1}[1] = \ldots = \mathbf{w}_{t_0}[1].$$

If $k = t_0$ then $\mathbf{w}_k[1] = \mathbf{w}_{t_0}[1]$ trivially holds. Thus,

$$\mathbf{w}_k = \begin{pmatrix} 1 - \frac{1-\alpha^2}{\lambda_{t_0}+1} \\ \zeta \end{pmatrix}$$

for some $\zeta \in \mathbb{R}$. Hence, setting $\alpha = \sqrt{1/2}$,

$$f_2(\mathbf{w}_k) = \frac{1}{2}\left(\left(1 - \frac{1-\alpha^2}{\lambda_{t_0}+1}\right)\sqrt{1-\alpha^2} + \alpha\zeta\right)^2 = \frac{1}{4}\left(1 - \frac{1}{2(\lambda_{t_0}+1)} + \zeta\right)^2,$$

and $f_1(\mathbf{w}_k) = \frac{1}{2}\zeta^2$. If $|\zeta| \geq \frac{1}{\sqrt{k}}$, we are done as $f_1$ is observed $k-1$ times (conditioned on $E_B$) and

$$\mathcal{L}_{1:k}(\mathbf{w}_k) \geq \frac{k-1}{k}f_1(w_k) = \frac{k-1}{2k}\zeta^2 = \Omega(1/k).$$

Otherwise, as $k \geq 9$, $\zeta > -1/3$, and (conditioned on $E_B$)

$$f_2(\mathbf{w}_k) \geq \frac{1}{4}(1/6)^2 = \frac{1}{144}.$$

Therefore, in this case,

$$\mathcal{L}_{1:k}(\mathbf{w}_k) \geq \frac{1}{k}f_2(\mathbf{w}_k) = \Omega(1/k).$$

So with probability at least $\Pr(E_B) \geq 1/4 \geq 1/10$, it holds that

$$\mathcal{L}_{1:k}(\mathbf{w}_k) = \Omega(1/k).$$

$\square$

# C Proofs of the reductions and their properties

**Recall Reduction 1 — Regularized Continual Regression $\Rightarrow$ Incremental GD.** Consider $M$ regression tasks $\{(\mathbf{X}_m, \mathbf{y}_m)\}_{m=1}^M$ seen in any ordering $\tau$. Then, each iteration $t \in [k]$ of regularized continual linear regression with a coefficient $\lambda_t > 0$ is equivalent to an IGD step on $f_r^{(t)}(\cdot; \tau_t)$ with step size $\eta_t > 0$, where $f_r^{(t)}(\mathbf{w}; m) \triangleq \frac{1}{2} \left\| \sqrt{\mathbf{A}_m^{(t)}} (\mathbf{w} - \mathbf{X}_m^+ \mathbf{y}_m) \right\|^2$ for some $\mathbf{A}_m^{(t)}$ depending on $\lambda_t, \eta_t$. That is, the iterates of Schemes 1 and 3 coincide.

*Proof of Reduction 1.* Each iterate of regularized continual regression is defined as

$$\mathbf{w}_t = \arg\min_{\mathbf{w}} \left( \frac{1}{2} \|\mathbf{X}_{\tau_t} \mathbf{w} - \mathbf{y}_{\tau_t}\|^2 + \frac{\lambda_t}{2} \|\mathbf{w} - \mathbf{w}_{t-1}\|^2 \right),$$

which admits the closed-form update:

$$\mathbf{w}_t = \left( \mathbf{X}_{\tau_t}^\top \mathbf{X}_{\tau_t} + \lambda_t \mathbf{I} \right)^{-1} \left( \mathbf{X}_{\tau_t}^\top \mathbf{y}_{\tau_t} + \lambda_t \mathbf{w}_{t-1} \right).$$

We define:

$$\mathbf{A}_m \triangleq \frac{1}{\eta_t} \left( \mathbf{I} - \lambda_t \left( \mathbf{X}_m^\top \mathbf{X}_m + \lambda_t \mathbf{I} \right)^{-1} \right), \qquad f_r^{(t)}(\mathbf{w}; m) \triangleq \frac{1}{2} \left\| \sqrt{\mathbf{A}_m} (\mathbf{w} - \mathbf{X}_m^+ \mathbf{y}_m) \right\|^2.$$

(For notational simplicity, we write $\mathbf{A}_m$ in place of the more precise $\mathbf{A}_m^{(t)}$.) Observe that:

$$\begin{aligned}
\eta_t \mathbf{A}_m &= \mathbf{I} - \lambda_t \left( \mathbf{X}_m^\top \mathbf{X}_m + \lambda_t \mathbf{I} \right)^{-1} \\
&= \left( \mathbf{X}_m^\top \mathbf{X}_m + \lambda_t \mathbf{I} \right) \left( \mathbf{X}_m^\top \mathbf{X}_m + \lambda_t \mathbf{I} \right)^{-1} - \lambda_t \left( \mathbf{X}_m^\top \mathbf{X}_m + \lambda_t \mathbf{I} \right)^{-1} \\
&= \mathbf{X}_m^\top \mathbf{X}_m \left( \mathbf{X}_m^\top \mathbf{X}_m + \lambda_t \mathbf{I} \right)^{-1}.
\end{aligned}$$

When we run IGD on $f_r^{(t)}$ with learning rate $\eta_t$, we get:

$$\begin{aligned}
\mathbf{w}_{t-1} - \eta_t \nabla f_r^{(t)}(\mathbf{w}_{t-1}; \tau_t) &= \mathbf{w}_{t-1} - \eta_t \mathbf{A}_{\tau_t} \left( \mathbf{w}_{t-1} - \mathbf{X}_{\tau_t}^+ \mathbf{y}_{\tau_t} \right) \\
&= \lambda_t \left( \mathbf{X}_{\tau_t}^\top \mathbf{X}_{\tau_t} + \lambda_t \mathbf{I} \right)^{-1} \mathbf{w}_{t-1} + \mathbf{X}_{\tau_t}^\top \mathbf{X}_{\tau_t} \left( \mathbf{X}_{\tau_t}^\top \mathbf{X}_{\tau_t} + \lambda_t \mathbf{I} \right)^{-1} \mathbf{X}_{\tau_t}^+ \mathbf{y}_{\tau_t} \\
&= \lambda_t \left( \mathbf{X}_{\tau_t}^\top \mathbf{X}_{\tau_t} + \lambda_t \mathbf{I} \right)^{-1} \mathbf{w}_{t-1} + \left( \mathbf{X}_{\tau_t}^\top \mathbf{X}_{\tau_t} + \lambda_t \mathbf{I} \right)^{-1} \mathbf{X}_{\tau_t}^\top \mathbf{y}_{\tau_t} \\
&= \left( \mathbf{X}_{\tau_t}^\top \mathbf{X}_{\tau_t} + \lambda_t \mathbf{I} \right)^{-1} \left( \lambda_t \mathbf{w}_{t-1} + \mathbf{X}_{\tau_t}^\top \mathbf{y}_{\tau_t} \right) = \mathbf{w}_t.
\end{aligned}$$

$\square$

**Recall Reduction 2 — Budgeted Continual Regression $\Rightarrow$ Incremental GD.** Consider $M$ regression tasks $\{(\mathbf{X}_m, \mathbf{y}_m)\}_{m=1}^M$ seen in any ordering $\tau$. Then, each iteration $t \in [k]$ of budgeted continual linear regression with $N_t \in \mathbb{N}$ inner steps of size $\gamma_t \in \left(0, 1/R^2\right)$ is equivalent to an IGD step on $f_b^{(t)}(\cdot; \tau_t)$ with step size $\eta_t > 0$, where $f_b^{(t)}(\mathbf{w}; m) \triangleq \frac{1}{2} \left\| \sqrt{\mathbf{A}_m^{(t)}} (\mathbf{w} - \mathbf{X}_m^+ \mathbf{y}_m) \right\|^2$ for some $\mathbf{A}_m^{(t)}$ depending on $N_t, \gamma_t, \eta_t$. That is, the iterates of Schemes 2 and 3 coincide.

*Proof of Reduction 2.* In budgeted continual regression, we apply $N_t$ steps of gradient descent with step size $\gamma_t$ to the loss $\frac{1}{2} \|\mathbf{X}_{\tau_t} \mathbf{w} - \mathbf{y}_{\tau_t}\|^2$. Let $\mathbf{w}^{(0)} \triangleq \mathbf{w}_{t-1}$. The inner iterates evolve as:

$$\mathbf{w}^{(s)} = \left( \mathbf{I} - \gamma_t \mathbf{X}_{\tau_t}^\top \mathbf{X}_{\tau_t} \right) \mathbf{w}^{(s-1)} + \gamma_t \mathbf{X}_{\tau_t}^\top \mathbf{y}_{\tau_t},$$

$$\mathbf{w}_t = \mathbf{w}^{(N_t)} = \left( \mathbf{I} - \gamma_t \mathbf{X}_{\tau_t}^\top \mathbf{X}_{\tau_t} \right)^{N_t} \mathbf{w}_{t-1} + \gamma_t \sum_{s=0}^{N_t-1} \left( \mathbf{I} - \gamma_t \mathbf{X}_{\tau_t}^\top \mathbf{X}_{\tau_t} \right)^s \mathbf{X}_{\tau_t}^\top \mathbf{y}_{\tau_t}.$$

We define:

$$\mathbf{A}_m \triangleq \frac{1}{\eta_t} \left( \mathbf{I} - \left( \mathbf{I} - \gamma_t \mathbf{X}_m^\top \mathbf{X}_m \right)^{N_t} \right), \qquad f_b^{(t)}(\mathbf{w}; m) \triangleq \frac{1}{2} \left\| \sqrt{\mathbf{A}_m} (\mathbf{w} - \mathbf{X}_m^+ \mathbf{y}_m) \right\|^2.$$

(For notational simplicity, we write $\mathbf{A}_m$ in place of the more precise $\mathbf{A}_m^{(t)}$.) To simplify the expression for the sum, consider the SVD $\mathbf{X}_{\tau_t} = \mathbf{U}\boldsymbol{\Sigma}\mathbf{V}^\top$ and observe:

$$\gamma_t \sum_{s=0}^{N_t-1} \left(\mathbf{I} - \gamma_t \mathbf{X}_{\tau_t}^\top \mathbf{X}_{\tau_t}\right)^s \mathbf{X}_{\tau_t}^\top \mathbf{y}_{\tau_t} = \mathbf{V} \sum_{s=0}^{N_t-1} \gamma_t \left(\mathbf{I} - \gamma_t \boldsymbol{\Sigma}^2\right)^s \boldsymbol{\Sigma}\mathbf{U}^\top \mathbf{y}_{\tau_t}$$

$$[\text{Geometric sum}] = \mathbf{V} \left(\mathbf{I} - \left(\mathbf{I} - \gamma_t \boldsymbol{\Sigma}^2\right)^{N_t}\right) \boldsymbol{\Sigma}^+ \mathbf{U}^\top \mathbf{y}_{\tau_t} = \left(\mathbf{I} - \left(\mathbf{I} - \gamma_t \mathbf{X}_{\tau_t}^\top \mathbf{X}_{\tau_t}\right)^{N_t}\right) \mathbf{X}_{\tau_t}^+ \mathbf{y}_{\tau_t}$$

$$= \eta_t \mathbf{A}_{\tau_t} \mathbf{X}_{\tau_t}^+ \mathbf{y}_{\tau_t}.$$

When we run IGD on $f_b^{(t)}$ with learning rate $\eta_t$. We have:

$$\mathbf{w}_{t-1} - \eta_t \nabla f_b^{(t)} \left(\mathbf{w}_{t-1}; \tau_t\right) = \mathbf{w}_{t-1} - \eta_t \mathbf{A}_{\tau_t} \left(\mathbf{w}_{t-1} - \mathbf{X}_{\tau_t}^+ \mathbf{y}_{\tau_t}\right)$$

$$= \left(\mathbf{I} - \eta_t \mathbf{A}_{\tau_t}\right) \mathbf{w}_{t-1} + \eta_t \mathbf{A}_{\tau_t} \mathbf{X}_{\tau_t}^+ \mathbf{y}_{\tau_t}$$

$$= \left(\mathbf{I} - \gamma_t \mathbf{X}_{\tau_t}^\top \mathbf{X}_{\tau_t}\right)^{N_t} \mathbf{w}_{t-1} + \gamma_t \sum_{s=0}^{N_t-1} \left(\mathbf{I} - \gamma_t \mathbf{X}_{\tau_t}^\top \mathbf{X}_{\tau_t}\right)^s \mathbf{X}_{\tau_t}^\top \mathbf{y}_{\tau_t} = \mathbf{w}_t.$$

$\square$

**Lemma C.1** (General reduction properties). *Recall $\mathcal{L}\left(\mathbf{w}; m\right) \triangleq \frac{1}{2}\|\mathbf{X}_m\mathbf{w} - \mathbf{y}_m\|^2$ and $R^2 \triangleq \max_{m'}\|\mathbf{X}_{m'}\|_2^2$. Let*

$$f^{(t)}\left(\mathbf{w}; m\right) \triangleq \frac{1}{2}\left\|\sqrt{\mathbf{A}_m}\left(\mathbf{w} - \mathbf{X}_m^+\mathbf{y}_m\right)\right\|^2 \quad \text{with} \quad \mathbf{A}_m = g\left(\mathbf{X}_m^\top\mathbf{X}_m\right),$$

*where $g : \mathbb{R} \to \mathbb{R}$ is applied spectrally (i.e., to each eigenvalue of $\mathbf{X}_m^\top\mathbf{X}_m$). Assume that $g$ is concave, non-decreasing on $[0, R^2]$, with $g(0) = 0$ and $g'(0) > 0$. Then:*

*(i) $f^{(t)}\left(\mathbf{w}; m\right)$ is $g\left(R^2\right)$-smooth,*

*(ii) and the following inequality holds:*

$$\frac{1}{g'(0)}f^{(t)}\left(\mathbf{w}; m\right) \leq \mathcal{L}\left(\mathbf{w}; m\right) - \min_{\mathbf{w}'}\mathcal{L}\left(\mathbf{w}'; m\right) \leq \frac{R^2}{g\left(R^2\right)}f^{(t)}\left(\mathbf{w}; m\right).$$

*Proof.* Let $\xi_i$ denote the $i$-th eigenvalue of $\mathbf{X}_m^\top\mathbf{X}_m$, and let $\xi_i' \triangleq g(\xi_i)$ be the corresponding eigenvalue of $\mathbf{A}_m$. By the concavity of $g$, for every $\xi_i \in [0, R^2]$,

$$g(\xi_i) \leq g'(0) \cdot \xi_i \quad \Rightarrow \quad \frac{1}{g'(0)}\xi_i' \leq \xi_i.$$

Hence, $\frac{1}{g'(0)}\mathbf{A}_m \preccurlyeq \mathbf{X}_m^\top\mathbf{X}_m$. By concavity and $g(0) = 0$, the chord from $0$ to $R^2$ lies below $g$:

$$g(\xi_i) \geq \frac{g(R^2)}{R^2} \cdot \xi_i \quad \Rightarrow \quad \xi_i \leq \frac{R^2}{g(R^2)} \cdot \xi_i',$$

so we obtain the matrix inequality: $\mathbf{X}_m^\top\mathbf{X}_m \preccurlyeq \frac{R^2}{g(R^2)}\mathbf{A}_m$.

Moreover, since $g$ is non-decreasing, $\xi_i' \leq g(R^2)$, and therefore all eigenvalues of $\mathbf{A}_m$ are upper bounded by $g(R^2)$, $\mathbf{A}_m \preccurlyeq g(R^2)\mathbf{I}$, implying that the Hessian $\nabla^2 f^{(t)}(\mathbf{w}; m) = \mathbf{A}_m$ satisfies smoothness with parameter $g(R^2)$.

Next, decompose the squared loss:

$$\mathcal{L}(\mathbf{w}; m) = \frac{1}{2}\|\mathbf{X}_m\mathbf{w} - \mathbf{y}_m\|^2 = \frac{1}{2}\left\|\mathbf{X}_m\left(\mathbf{w} - \mathbf{X}_m^+\mathbf{y}_m\right) + \left(\mathbf{X}_m\mathbf{X}_m^+ - \mathbf{I}\right)\mathbf{y}_m\right\|^2$$

$$[\text{Orthogonality}] = \frac{1}{2}\left(\left\|\mathbf{X}_m\left(\mathbf{w} - \mathbf{X}_m^+\mathbf{y}_m\right)\right\|^2 + \left\|\left(\mathbf{X}_m\mathbf{X}_m^+ - \mathbf{I}\right)\mathbf{y}_m\right\|^2\right).$$

where the two terms are orthogonal since $\mathbf{X}_m\left(\mathbf{w} - \mathbf{X}_m^+\mathbf{y}_m\right) \in \text{range}(\mathbf{X}_m)$ and $\left(\mathbf{X}_m\mathbf{X}_m^+ - \mathbf{I}\right)\mathbf{y}_m \in \ker(\mathbf{X}_m^\top)$.

The minimum loss is attained at $\mathbf{X}_m^+ \mathbf{y}_m$, yielding: $\min_{\mathbf{w}'} \mathcal{L}(\mathbf{w}'; m) = \frac{1}{2} \left\| (\mathbf{X}_m \mathbf{X}_m^+ - \mathbf{I}) \mathbf{y}_m \right\|^2$. Thus, the excess loss becomes:

$$\mathcal{L}(\mathbf{w}; m) - \min_{\mathbf{w}'} \mathcal{L}(\mathbf{w}'; m) = \frac{1}{2} \left\| \mathbf{X}_m \left( \mathbf{w} - \mathbf{X}_m^+ \mathbf{y}_m \right) \right\|^2 = \frac{1}{2} \left( \mathbf{w} - \mathbf{X}_m^+ \mathbf{y}_m \right)^\top \mathbf{X}_m^\top \mathbf{X}_m \left( \mathbf{w} - \mathbf{X}_m^+ \mathbf{y}_m \right).$$

Meanwhile, $f^{(t)}(\mathbf{w}; m) = \frac{1}{2} \left( \mathbf{w} - \mathbf{X}_m^+ \mathbf{y}_m \right)^\top \mathbf{A}_m \left( \mathbf{w} - \mathbf{X}_m^+ \mathbf{y}_m \right)$.

By the sandwich inequality $\frac{1}{g'(0)} \mathbf{A}_m \preccurlyeq \mathbf{X}_m^\top \mathbf{X}_m \preccurlyeq \frac{R^2}{g(R^2)} \mathbf{A}_m$, we conclude:

$$\frac{1}{g'(0)} f^{(t)}(\mathbf{w}; m) \leq \mathcal{L}(\mathbf{w}; m) - \min_{\mathbf{w}'} \mathcal{L}(\mathbf{w}'; m) \leq \frac{R^2}{g(R^2)} f^{(t)}(\mathbf{w}; m). \qquad \square$$

**Recall Lemma 3.1 — properties of the IGD objectives.** For $t \in [k]$, define $f_r^{(t)}, f_b^{(t)}$ as in Reductions 1 and 2, and recall the data radius $R \triangleq \max_{m \in [M]} \|\mathbf{X}_m\|_2$.

(i) $f_r^{(t)}, f_b^{(t)}$ are both convex and $\beta$-smooth for $\beta_r^{(t)} \triangleq \frac{1}{\eta_t} \frac{R^2}{R^2 + \lambda_t}$, $\beta_b^{(t)} \triangleq \frac{1}{\eta_t} \left( 1 - (1 - \gamma_t R^2)^{N_t} \right)$.

(ii) Both functions bound the "excess" loss from both sides, *i.e.*, $\forall \mathbf{w} \in \mathbb{R}^d, \forall t \in [k], \forall m \in [m]$,

$$\lambda_t \eta_t \cdot f_r^{(t)}(\mathbf{w}; m) \leq \mathcal{L}(\mathbf{w}; m) - \min_{\mathbf{w}'} \mathcal{L}(\mathbf{w}'; m) \leq \frac{R^2}{\beta_r^{(t)}} \cdot f_r^{(t)}(\mathbf{w}; m),$$

$$\frac{\eta_t}{\gamma_t N_t} \cdot f_b^{(t)}(\mathbf{w}; m) \leq \mathcal{L}(\mathbf{w}; m) - \min_{\mathbf{w}'} \mathcal{L}(\mathbf{w}'; m) \leq \frac{R^2}{\beta_b^{(t)}} \cdot f_b^{(t)}(\mathbf{w}; m).$$

(iii) Finally, when the tasks are jointly realizable (see Assumption 4.1), the same $\mathbf{w}_\star$ minimizes all surrogate objectives simultaneously. That is,

$$\mathcal{L}(\mathbf{w}_\star; m) = f_r^{(t)}(\mathbf{w}_\star; m) = f_b^{(t)}(\mathbf{w}_\star; m) = 0, \quad \forall t \in [k], \forall m \in [M].$$

*Proof of Lemma 3.1.* Recall the definitions of the IGD objectives:

$$f_r^{(t)}(\mathbf{w}; m) \triangleq \frac{1}{2} \left\| \sqrt{g_r(\mathbf{X}_m^\top \mathbf{X}_m)} \left( \mathbf{w} - \mathbf{X}_m^+ \mathbf{y}_m \right) \right\|^2,$$

$$f_b^{(t)}(\mathbf{w}; m) \triangleq \frac{1}{2} \left\| \sqrt{g_b(\mathbf{X}_m^\top \mathbf{X}_m)} \left( \mathbf{w} - \mathbf{X}_m^+ \mathbf{y}_m \right) \right\|^2,$$

where the functions $g_r, g_b : \mathbb{R} \to \mathbb{R}$ are applied spectrally (i.e., to the eigenvalues of $\mathbf{X}_m^\top \mathbf{X}_m$), and are defined as:

$$g_r(\xi) \triangleq \frac{1}{\eta_t} \left( 1 - \frac{\lambda_t}{\xi + \lambda_t} \right), \qquad g_b(\xi) \triangleq \frac{1}{\eta_t} \left( 1 - (1 - \gamma_t \xi)^{N_t} \right).$$

Note that both $f_r^{(t)}$ and $f_b^{(t)}$ are standard quadratic forms and hence convex in $\mathbf{w}$.

We verify that $g_r$ and $g_b$ satisfy the assumptions of Lemma C.1 on the domain $\xi \in [0, R^2]$:

- $g_r$ is differentiable with

$$g_r'(\xi) = \frac{\lambda_t}{\eta_t (\xi + \lambda_t)^2} \geq 0, \qquad g_r''(\xi) = -\frac{2\lambda_t}{\eta_t (\xi + \lambda_t)^3} \leq 0,$$

  so $g_r$ is non-decreasing and concave.

- $g_b$ is differentiable with

$$g_b'(\xi) = \frac{N_t \gamma_t}{\eta_t} (1 - \gamma_t \xi)^{N_t - 1} \geq 0, \qquad g_b''(\xi) = -\frac{N_t (N_t - 1) \gamma_t^2}{\eta_t} (1 - \gamma_t \xi)^{N_t - 2} \leq 0,$$

  Thus, $g_b$ is also non-decreasing and concave.

In addition, we note:

$$g_r(0) = 0, \quad g'_r(0) = \frac{1}{\eta_t \lambda_t} > 0, \qquad g_b(0) = 0, \quad g'_b(0) = \frac{N_t \gamma_t}{\eta_t} > 0,$$

and we compute the smoothness constants:

$$g_r(R^2) = \frac{R^2}{\eta_t(R^2 + \lambda_t)} = \frac{1}{\beta_r^{(t)}}, \qquad g_b(R^2) = \frac{1}{\eta_t}\left(1 - (1 - \gamma_t R^2)^{N_t}\right) = \frac{1}{\beta_b^{(t)}}.$$

Hence, by Lemma C.1, both $f_r^{(t)}$ and $f_b^{(t)}$ are $\beta^{(t)}$-smooth with the claimed parameters $\beta_r^{(t)}, \beta_b^{(t)}$, and they satisfy the two-sided bounds:

$$\frac{1}{g'_r(0)} f_r^{(t)}(\mathbf{w}; m) \le \mathcal{L}(\mathbf{w}; m) - \min_{\mathbf{w}'} \mathcal{L}(\mathbf{w}'; m) \le \frac{R^2}{g_r(R^2)} f_r^{(t)}(\mathbf{w}; m),$$

$$\frac{1}{g'_b(0)} f_b^{(t)}(\mathbf{w}; m) \le \mathcal{L}(\mathbf{w}; m) - \min_{\mathbf{w}'} \mathcal{L}(\mathbf{w}'; m) \le \frac{R^2}{g_b(R^2)} f_b^{(t)}(\mathbf{w}; m).$$

Substituting in $g'_r(0)$ and $g'_b(0)$ yields the bounds stated in part (ii).

Finally, for part (iii), assume the tasks satisfy joint realizability (Assumption 4.1), meaning that for some common minimizer $\mathbf{w}_\star$,

$$\mathcal{L}(\mathbf{w}_\star; m) = \min_{\mathbf{w}'} \mathcal{L}(\mathbf{w}'; m), \quad \forall m.$$

Then by the lower bounds in part (ii), both $f_r^{(t)}(\mathbf{w}_\star; m) = 0$ and $f_b^{(t)}(\mathbf{w}_\star; m) = 0$ for all $t, m$, completing the proof. $\qquad\square$

# D  Proofs for fixed regularization strength

**Recall Lemma 4.3 — rates for fixed regularization strength.** Assume a random with-replacement ordering over jointly realizable tasks. Then, for each of Schemes 1 and 2, the expected loss after $k \geq 1$ iterations is upper bounded as:

(i) **Fixed coefficient:** For Scheme 1 with a regularization coefficient $\lambda > 0$,

$$\mathbb{E}_\tau \mathcal{L}(\mathbf{w}_k) \leq \frac{e \left\| \mathbf{w}_0 - \mathbf{w}_\star \right\|^2 R^2}{2 \cdot \frac{R^2}{R^2 + \lambda} \cdot \left( 2 - \frac{R^2}{R^2 + \lambda} \right) \cdot k^{1 - \frac{R^2}{R^2 + \lambda} \left( 1 - \frac{R^2}{4(R^2 + \lambda)} \right)}} \; ;$$

(ii) **Fixed budget:** For Scheme 2 with step size $\gamma \in (0, 1/R^2)$ and budget $N \in \mathbb{N}$,

$$\mathbb{E}_\tau \mathcal{L}(\mathbf{w}_k) \leq \frac{e \left\| \mathbf{w}_0 - \mathbf{w}_\star \right\|^2 R^2}{2 \cdot \left( 1 - (1 - \gamma R^2)^{2N} \right) \cdot k^{1 - (1 - (1 - \gamma R^2)^N) \left( 1 - \frac{1 - (1 - \gamma R^2)^N}{4} \right)}} \; .$$

*Proof of Lemma 4.3.* From Reductions 1 and 2, the iterates of Schemes 1 and 2 are equivalent to those of IGD (Scheme 3) applied to the respective surrogate objectives $f_r^{(t)}$ and $f_b^{(t)}$. When $\eta, \lambda, \gamma, N$ are fixed, the functions $f_r^{(t)}, f_b^{(t)}$ do not depend on $t$, and under a random ordering with replacement, the update rule becomes standard SGD.

By Lemma 3.1, the surrogates $f_r^{(t)}$ and $f_b^{(t)}$ are jointly realizable whenever the original losses are, and hence satisfy the assumptions of the following result from Evron et al. [14].

> *Rephrased Theorem 5.1 of Evron et al. [14]:* Let $\bar{f}(\mathbf{w}) \triangleq \frac{1}{M} \sum_{m=1}^{M} f(\mathbf{w}; m)$, where each $f(\mathbf{w}; m) \triangleq \frac{1}{2} \left\| \tilde{\mathbf{A}}_m \mathbf{w} - \tilde{\mathbf{b}}_m \right\|^2$ is $\beta$-smooth, and assume realizability: $\bar{f}(\mathbf{w}_\star) = 0$ for some $\mathbf{w}_\star$. Then for any initialization $\mathbf{w}_0$ and step size $\eta \in \left( 0, \frac{2}{\beta} \right)$, SGD with replacement satisfies:
> $$\mathbb{E}_\tau \bar{f}(\mathbf{w}_k) \leq \frac{e \left\| \mathbf{w}_0 - \mathbf{w}_\star \right\|^2}{2\eta(2 - \eta\beta) \cdot k^{1 - \eta\beta(1 - \eta\beta/4)}}.$$

We now instantiate this result for each setting:

*(i) Fixed Regularization.* For Scheme 1, the surrogate $f_r^{(t)}$ is $\beta_r$-smooth with

$$\beta_r \triangleq \frac{1}{\eta} \cdot \frac{R^2}{R^2 + \lambda}, \quad \text{which implies} \quad \eta = \frac{1}{\beta_r} \cdot \frac{R^2}{R^2 + \lambda} < \frac{2}{\beta_r}.$$

The loss is upper bounded by the surrogate:

$$\mathcal{L}(\mathbf{w}_k) \leq \frac{R^2}{\beta_r} \cdot \bar{f}_r(\mathbf{w}_k),$$

which gives:

$$\mathbb{E}_\tau \mathcal{L}(\mathbf{w}_k) \leq \frac{R^2}{\beta_r} \cdot \mathbb{E}_\tau \bar{f}_r(\mathbf{w}_k) \leq \frac{e \left\| \mathbf{w}_0 - \mathbf{w}_\star \right\|^2 R^2}{2\eta\beta_r(2 - \eta\beta_r) \cdot k^{1 - \eta\beta_r(1 - \eta\beta_r/4)}}.$$

Substituting $\beta_r = \frac{1}{\eta} \cdot \frac{R^2}{R^2 + \lambda}$ gives:

$$\mathbb{E}_\tau \mathcal{L}(\mathbf{w}_k) \leq \frac{e \left\| \mathbf{w}_0 - \mathbf{w}_\star \right\|^2 R^2}{2 \cdot \frac{R^2}{R^2 + \lambda} \cdot \left( 2 - \frac{R^2}{R^2 + \lambda} \right) \cdot k^{1 - \frac{R^2}{R^2 + \lambda} \left( 1 - \frac{R^2}{4(R^2 + \lambda)} \right)}}.$$

*(ii) Fixed Budget.* For Scheme 2, the surrogate $f_b^{(t)}$ is $\beta_b$-smooth with

$$\beta_b \triangleq \frac{1}{\eta} \cdot \left( 1 - (1 - \gamma R^2)^N \right), \quad \text{so that} \quad \eta = \frac{1}{\beta_b} \cdot \left( 1 - (1 - \gamma R^2)^N \right) < \frac{2}{\beta_b}.$$

As before, we have:

$$\mathbb{E}_\tau \mathcal{L}(\mathbf{w}_k) \le \frac{R^2}{\beta_b} \cdot \mathbb{E}_\tau \bar{f}_b(\mathbf{w}_k) \le \frac{e \left\| \mathbf{w}_0 - \mathbf{w}_\star \right\|^2 R^2}{2\eta\beta_b(2 - \eta\beta_b) \cdot k^{1-\eta\beta_b(1-\eta\beta_b/4)}}.$$

Substituting $\beta_b = \frac{1}{\eta} \cdot \left(1 - (1 - \gamma R^2)^N\right)$ yields:

$$\mathbb{E}_\tau \mathcal{L}(\mathbf{w}_k) \le \frac{e \left\| \mathbf{w}_0 - \mathbf{w}_\star \right\|^2 R^2}{2 \cdot \left(1 - (1 - \gamma R^2)^{2N}\right) \cdot k^{1-(1-(1-\gamma R^2)^N)\left(1 - \frac{1-(1-\gamma R^2)^N}{4}\right)}}.$$

This completes the proof.

To extend this result to the without-replacement case (see Remark 4.5), we can simply invoke the without-replacement extension of Theorem 5.1 in Evron et al. [14]. □

**Recall Corollary 4.4 — near-optimal rates via fixed regularization strength.** Assume a random with-replacement ordering over jointly realizable tasks. When the regularization strengths in Lemma 4.3 are set as follows:

(i) **Fixed coefficient:** For Scheme 1, set regularization coefficient $\lambda \triangleq R^2(\ln k - 1)$;

(ii) **Fixed budget:** For Scheme 2, choose step size $\gamma \in (0, 1/R^2)$ and set budget $N \triangleq \frac{\ln\left(1 - \frac{1}{\ln k}\right)}{\ln(1-\gamma R^2)}$;

Then, under either Scheme 1 or Scheme 2, the expected loss after $k \ge 2$ iterations is bounded as:

$$\mathbb{E}_\tau \mathcal{L}\left(\mathbf{w}_k\right) \le \frac{5 \left\| \mathbf{w}_0 - \mathbf{w}_\star \right\|^2 R^2 \ln k}{k}.$$

*Proof of Corollary 4.4.* We apply the general loss bound from Lemma 4.3, which holds for both fixed-regularization and fixed-budget variants:

$$\mathbb{E}_\tau \mathcal{L}(\mathbf{w}_k) \le \frac{e \left\| \mathbf{w}_0 - \mathbf{w}_\star \right\|^2 R^2}{2\eta\beta \left(2 - \eta\beta\right) \cdot k^{1-\eta\beta(1-\eta\beta/4)}}.$$

Now plug in the parameter settings from the statement of the lemma.

*(i) Fixed Regularization.* Set $\lambda \triangleq R^2(\ln k - 1)$. Then:

$$\beta_r = \frac{1}{\eta} \cdot \frac{R^2}{R^2 + \lambda} = \frac{1}{\eta} \cdot \frac{R^2}{R^2 + R^2(\ln k - 1)} = \frac{1}{\eta} \cdot \frac{1}{\ln k} \quad \Rightarrow \quad \eta\beta_r = \frac{1}{\ln k}.$$

*(ii) Fixed Budget.* Set $N \triangleq \frac{\ln\left(1 - \frac{1}{\ln k}\right)}{\ln(1-\gamma R^2)}$. Then:

$$(1 - \gamma R^2)^N = 1 - \frac{1}{\ln k} \quad \Rightarrow \quad \beta_b = \frac{1}{\eta} \cdot \left(1 - (1 - \gamma R^2)^N\right) = \frac{1}{\eta} \cdot \frac{1}{\ln k} \quad \Rightarrow \quad \eta\beta_b = \frac{1}{\ln k}.$$

In both cases, we have $\eta\beta = \frac{1}{\ln k}$. Substituting into the loss bound:

$$\begin{aligned}
\mathbb{E}_\tau \mathcal{L}(\mathbf{w}_k) &\le \frac{e \left\| \mathbf{w}_0 - \mathbf{w}_\star \right\|^2 R^2}{\frac{2}{\ln k} \cdot \left(2 - \frac{1}{\ln k}\right) \cdot k^{1 - \frac{1}{\ln k}\left(1 - \frac{1}{4\ln k}\right)}} \\
&= \left\| \mathbf{w}_0 - \mathbf{w}_\star \right\|^2 R^2 \cdot \frac{e \ln k}{2\left(2 - \frac{1}{\ln k}\right)} \cdot \frac{1}{k} \cdot k^{\frac{1}{\ln k} - \frac{1}{4(\ln k)^2}} \\
&= \frac{\left\| \mathbf{w}_0 - \mathbf{w}_\star \right\|^2 R^2 \ln k}{k} \cdot \frac{e^{2 - \frac{1}{4\ln k}}}{2\left(2 - \frac{1}{\ln k}\right)}.
\end{aligned}$$

Since $e^{2 - \frac{1}{4\ln k}} / \left(2 - \frac{1}{\ln k}\right) \le 5$ for all $k \ge 2$, we conclude:

$$\mathbb{E}_\tau \mathcal{L}(\mathbf{w}_k) \le \frac{5 \left\| \mathbf{w}_0 - \mathbf{w}_\star \right\|^2 R^2 \ln k}{k}.$$

□

# E Proofs for scheduled regularization strength

**Recall Theorem 4.6 — optimal rates for increasing regularization.** Assume a random with-replacement ordering over jointly realizable tasks. Consider either Scheme 1 or Scheme 2 with the following time-dependent schedules:

(i) **Scheduled coefficient:** For Scheme 1, set regularization coefficient $\lambda_t = \dfrac{13R^2}{3} \cdot \dfrac{k+1}{k-t+2}$;

(ii) **Scheduled budget:**

For Scheme 2, choose step sizes $\gamma_t \in (0, 1/R^2)$ and set budget $N_t = \dfrac{3}{13\gamma_t R^2} \cdot \dfrac{k-t+2}{k+1}$;

Then, under either Scheme 1 or Scheme 2, the expected loss after $k \geq 2$ iterations is bounded as:

$$\mathbb{E}_\tau \mathcal{L}(\mathbf{w}_k) \leq \frac{20 \left\| \mathbf{w}_0 - \mathbf{w}_\star \right\|^2 R^2}{k+1}.$$

*Proof of Theorem 4.6.* We apply Lemma 4.7 with the original loss $f(\mathbf{w}; m) = \mathcal{L}(\mathbf{w}; m)$ and surrogates $f^{(t)}(\mathbf{w}; m) = f_r^{(t)}(\mathbf{w}; m)$ or $f_b^{(t)}(\mathbf{w}; m)$, defined in Reductions 1 and 2.

*Smoothness and convexity.* From Lemma 3.1, both surrogates are convex. Their smoothness constants are:

$$\beta_r^{(t)} = \frac{1}{\eta_t} \cdot \frac{R^2}{R^2 + \lambda_t}, \qquad \beta_b^{(t)} = \frac{1}{\eta_t} \left( 1 - \left( 1 - \gamma_t R^2 \right)^{N_t} \right).$$

**Regularized:** Setting $\lambda_t = 1/\eta_t$ gives

$$\beta_r^{(t)} = \frac{R^2}{\eta_t R^2 + 1} \leq R^2.$$

**Budgeted:** With $N_t = \eta_t/\gamma_t$, we get $\frac{\eta_t R^2}{N_t} = \gamma_t R^2 \in (0, 1)$. Using $(1-x)^n \geq 1 - nx$, we obtain:

$$\beta_b^{(t)} \leq \frac{1}{\eta_t}(1 - (1 - \eta_t R^2)) = R^2.$$

Thus, both surrogates are $R^2$-smooth, matching the smoothness of the loss $\mathcal{L}(\cdot; m)$ and satisfying condition (i) of Lemma 4.7.

*Joint realizability.* From Lemma 3.1, if the original tasks are jointly realizable, then so are the surrogates:

$$f_r^{(t)}(\mathbf{w}_\star; m) = f_b^{(t)}(\mathbf{w}_\star; m) = \mathcal{L}(\mathbf{w}_\star; m) = 0, \quad \forall t \in [k], \ m \in [M],$$

so condition (iii) of Lemma 4.7 is satisfied.

*Two-sided bounds.* We verify condition (ii) of Lemma 4.7 using the two-sided inequalities from Lemma 3.1:

$$\lambda_t \eta_t \cdot f_r^{(t)}(\mathbf{w}; m) \ \leq \ \mathcal{L}(\mathbf{w}; m) - \min_{\mathbf{w}'} \mathcal{L}(\mathbf{w}'; m) \ \leq \ \frac{R^2}{\beta_r^{(t)}} \cdot f_r^{(t)}(\mathbf{w}; m),$$

$$\frac{\eta_t}{\gamma_t N_t} \cdot f_b^{(t)}(\mathbf{w}; m) \ \leq \ \mathcal{L}(\mathbf{w}; m) - \min_{\mathbf{w}'} \mathcal{L}(\mathbf{w}'; m) \ \leq \ \frac{R^2}{\beta_b^{(t)}} \cdot f_b^{(t)}(\mathbf{w}; m).$$

By our choice of $\lambda_t = 1/\eta_t$ and $N_t = \eta_t/\gamma_t$, we have $\lambda_t \eta_t = \frac{\eta_t}{\gamma_t N_t} = 1$, so the lower bounds reduce to

$$f_r^{(t)}(\mathbf{w}; m) \leq \mathcal{L}(\mathbf{w}; m), \qquad f_b^{(t)}(\mathbf{w}; m) \leq \mathcal{L}(\mathbf{w}; m).$$

Now set $\nu_t \triangleq \eta_t$. To satisfy the upper bound $\mathcal{L}(\mathbf{w}; m) \leq (1 + \nu_t \beta) \cdot f^{(t)}(\mathbf{w}; m)$, it suffices to show

$$\frac{R^2}{\beta^{(t)}} \leq 1 + \eta_t R^2.$$

**Regularized:** $\beta_r^{(t)} = \frac{1}{\eta_t} \cdot \frac{R^2}{R^2 + \lambda_t} = \frac{R^2}{\eta_t R^2 + 1} \Rightarrow \frac{R^2}{\beta_r^{(t)}} = 1 + \eta_t R^2.$

**Budgeted:** With $\gamma_t R^2 = \frac{\eta_t R^2}{N_t} \in (0, 1)$, and using $\left(1 - \frac{x}{n}\right)^n \le e^{-x} \le \frac{1}{1+x}$ for $x \in (0, 1)$, we get:

$$\frac{R^2}{\beta_b^{(t)}} = \frac{R^2}{\frac{1}{\eta_t}\left(1 - \left(1 - \frac{\eta_t R^2}{N_t}\right)^{N_t}\right)} \le \frac{R^2}{\frac{1}{\eta_t}\left(1 - \frac{1}{1+\eta_t R^2}\right)} = 1 + \eta_t R^2.$$

Hence, both the lower and upper bounds hold, and condition (ii) is satisfied.

Setting the learning rate schedule to:

$$\eta = \frac{3}{13R^2}, \quad \text{and} \quad \eta_t = \eta \cdot \frac{k - t + 2}{k}.$$

Applying Lemma 4.7 yields:

$$\mathbb{E}_\tau \mathcal{L}(\mathbf{w}_k) = \mathbb{E}_\tau f(\mathbf{w}_k) \le \frac{20 \left\| \mathbf{w}_0 - \mathbf{w}_\star \right\|^2 R^2}{k + 1}.$$

$\square$

### E.1 Proof of Lemma 4.7

In this section, we provide the proof of our main lemma establishing the guarantees of time varying SGD. In order to better align with conventions in the optimization literature from which our techniques draw upon, we adopt different indexing for the SGD iterates throughout this section. Below, we restate the lemma with the alternative indexing scheme; the original Lemma 4.7 follows immediately by a simple shift of $k + 1 \to k$ and $1 \to 0$ in the indexes of the iterates $\mathbf{w}_t$.

**Lemma E.1** (Restatement of Lemma 4.7 with alternative indexing). *Assume $\tau$ is a random with-replacement ordering over $M$ jointly realizable convex and $\beta$-smooth loss functions $f(\cdot; m)\colon \mathbb{R}^d \to \mathbb{R}$. Define the average loss $f(\mathbf{w}) \triangleq \mathbb{E}_{m \sim \tau} f(\mathbf{w}; m)$. Let $k \geq 2$, and suppose $\left\{ f^{(t)}(\cdot; m) \mid t \in [k], m \in [M] \right\}$ for $t \in [k]$ are time-varying surrogate losses that satisfy:*

*(i) Smoothness and convexity: $f^{(t)}(\cdot; m)$ are $\beta$-smooth and convex for all $m \in [M], t \in [k]$;*

*(ii) There exists a weight sequence $\nu_1, \dots, \nu_k$ such that for all $m \in [M], t \in [k], \mathbf{w} \in \mathbb{R}^d$:*

$$f^{(t)}(\mathbf{w}; m) - f^{(t)}(\mathbf{w}_\star; m) \leq f(\mathbf{w}; m) - f(\mathbf{w}_\star; m) \leq (1 + \nu_t \beta)(f^{(t)}(\mathbf{w}; m) - f^{(t)}(\mathbf{w}_\star; m));$$

*(iii) Joint realizability:*

$$\mathbf{w}_\star \in \cap_{t \in [k]} \cap_{m \in [M]} \arg\min_{\mathbf{w}} f^{(t)}(\mathbf{w}; m); \quad \forall m \in [M], t \in [k], f^{(t)}(\mathbf{w}_\star; m) = f(\mathbf{w}_\star; m).$$

*Then, for any initialization $\mathbf{w}_1 \in \mathbb{R}^d$, the SGD updates:*

$$t = 1, \dots, k : \quad \mathbf{w}_{t+1} = \mathbf{w}_t - \eta_t \nabla f^{(t)}(\mathbf{w}_t; \tau_t)$$

*with a step size schedule that satisfies $\nu_t \leq \eta_t = \eta \left( \frac{(k+1) - t + 1}{k+1} \right) \forall t \in [k]$ for some $\eta \leq 3/(13\beta)$, guarantees the following **expected loss bound**:*

$$\mathbb{E} f(\mathbf{w}_{k+1}) - f(\mathbf{w}_\star) \leq \frac{9}{2\eta(k+1)} \|\mathbf{w}_1 - \mathbf{w}_\star\|^2 .$$

*In particular, for $\eta = \frac{3}{13\beta}$ we obtain*

$$\mathbb{E} f(\mathbf{w}_{k+1}) - f(\mathbf{w}_\star) \leq \frac{20\beta \|\mathbf{w}_1 - \mathbf{w}_\star\|^2}{k+1} .$$

*Furthermore, we also obtain the following **seen-task loss bound**:*

$$\mathbb{E} \left[ \frac{1}{k} \sum_{t=1}^{k} f(\mathbf{w}_{k+1}; \tau_t) - f(\mathbf{w}_\star; \tau_t) \right] \leq \frac{20}{\eta(k+1)} \|\mathbf{w}_1 - \mathbf{w}_\star\|^2 .$$

*In particular, for $\eta = \frac{3}{13\beta}$ we obtain*

$$\mathbb{E} \left[ \frac{1}{k} \sum_{t=1}^{k} f(\mathbf{w}_{k+1}; \tau_t) - f(\mathbf{w}_\star; \tau_t) \right] \leq \frac{87\beta \|\mathbf{w}_1 - \mathbf{w}_\star\|^2}{k+1} .$$

To prove the lemma above, we begin with a number of preliminary results. The next theorem provides an extension of [46] for our "relaxed SGD" setting that accommodates time varying distributions of functions.

**Theorem E.2.** *Let $J \geq 2$, and assume $\tau : [J] \to [M]$ is a random with-replacement ordering over $M$ jointly realizable convex and $\beta$-smooth loss functions $f(\cdot; m) \colon \mathbb{R}^d \to \mathbb{R}$. and suppose $\left\{ f^{(t)}(\cdot; m) \mid t \in [J], m \in [M] \right\}$ for $t \in [J]$ are time-varying surrogate losses for which there exists a weight sequence $\nu_1, \dots, \nu_J$ that satisfies, for all $m \in [M], t \in [J], \mathbf{w} \in \mathbb{R}^d$:*

$$f^{(t)}(\mathbf{w}; m) - f^{(t)}(\mathbf{w}_\star; m) \leq f(\mathbf{w}; m) - f(\mathbf{w}_\star; m) \leq (1 + \nu_t \beta) \left( f^{(t)}(\mathbf{w}; m) - f^{(t)}(\mathbf{w}_\star; m) \right).$$

*Then, for any initialization $\mathbf{w}_1 \in \mathbb{R}^d$ and step size sequence $\eta_1, \dots, \eta_J$, as long as $\forall t \in [J] : \eta_t \geq \nu_t$, the SGD updates:*

$$\mathbf{w}_{t+1} = \mathbf{w}_t - \eta_t \nabla f^{(t)}(\mathbf{w}_t; \tau_t), \tag{1}$$

*guarantee that for any $\mathbf{w}_\star \in \mathbb{R}^d$, and weight sequence $0 < v_0 \leq v_1 \leq \cdots \leq v_J$:*

$$\sum_{t=1}^{J} c_t \mathbb{E}\left[ \bar{f}^{(t)}(\mathbf{w}_t) - \bar{f}^{(t)}(\mathbf{w}_\star) \right] \leq \frac{v_0^2}{2} \|\mathbf{w}_1 - \mathbf{w}_\star\|^2 + \frac{1}{2} \sum_{t=1}^{J} \eta_t^2 v_t^2 \mathbb{E} \left\| \nabla f^{(t)}(\mathbf{w}_t; \tau_t) \right\|^2,$$

*where $c_t \triangleq \eta_t v_t^2 - (1 - \eta_t \beta)(v_t - v_{t-1}) \sum_{s=t}^{J} \eta_s v_s$, and $\bar{f}^{(t)}(\mathbf{w}) \triangleq \mathbb{E}_{m \sim \mathrm{Unif}[M]} f^{(t)}(\mathbf{w}; m)$.*

*Proof.* Define $\mathbf{z}_1, \dots, \mathbf{z}_J$ recursively by $\mathbf{z}_0 = \mathbf{w}_\star$ and for $t \geq 1$:

$$\mathbf{z}_t = \frac{v_{t-1}}{v_t} \mathbf{z}_{t-1} + \left( 1 - \frac{v_{t-1}}{v_t} \right) \mathbf{w}_t.$$

Denote $\mathbf{g}_t \triangleq \nabla f^{(t)}(\mathbf{w}_t; \tau_t)$ and observe,

$$\begin{aligned}
\|\mathbf{w}_{t+1} - \mathbf{z}_{t+1}\|^2 &= \frac{v_t^2}{v_{t+1}^2} \|\mathbf{w}_{t+1} - \mathbf{z}_t\|^2 \\
&= \frac{v_t^2}{v_{t+1}^2} \|\mathbf{w}_t - \eta_t \mathbf{g}_t - \mathbf{z}_t\|^2 \\
&= \frac{v_t^2}{v_{t+1}^2} \left( \|\mathbf{w}_t - \mathbf{z}_t\|^2 - 2\eta_t \langle \mathbf{g}_t, \mathbf{w}_t - \mathbf{z}_t \rangle + \eta_t^2 \|g_t\|^2 \right),
\end{aligned}$$

thus, rearranging we obtain

$$2v_t^2 \eta_t \langle \mathbf{g}_t, \mathbf{w}_t - \mathbf{z}_t \rangle = v_t^2 \|\mathbf{w}_t - \mathbf{z}_t\|^2 - v_{t+1}^2 \|\mathbf{w}_{t+1} - \mathbf{z}_{t+1}\|^2 + v_t^2 \eta_t^2 \|\mathbf{g}_t\|^2.$$

Summing over $t = 1, \dots, J$ yields

$$\sum_{t=1}^{J} v_t^2 \eta_t \langle \mathbf{g}_t, \mathbf{w}_t - \mathbf{z}_t \rangle \leq \frac{1}{2} v_0^2 \|\mathbf{w}_1 - \mathbf{w}_\star\|^2 + \frac{1}{2} \sum_{t=1}^{J} v_t^2 \eta_t^2 \|\mathbf{g}_t\|^2,$$

where we used that,

$$\|\mathbf{w}_1 - \mathbf{z}_1\| = \frac{v_0}{v_1} \|\mathbf{w}_1 - \mathbf{z}_0\| = \frac{v_0}{v_1} \|\mathbf{w}_1 - \mathbf{w}_\star\|.$$

Next, by convexity of $\bar{f}^{(t)}$ and the fact that $\mathbf{w}_t, \mathbf{z}_t$ are independent of $\tau_t$, conditioned on $\tau_1, \dots, \tau_{t-1}$:

$$\begin{aligned}
\mathbb{E}_{\tau_t} \langle \mathbf{g}_t, \mathbf{w}_t - \mathbf{z}_t \rangle &= \left\langle \mathbb{E}_{\tau_t} [\nabla f^{(t)}(\mathbf{w}_t; \tau_t)], \mathbf{w}_t - \mathbf{z}_t \right\rangle \\
&= \left\langle \nabla \bar{f}^{(t)}(\mathbf{w}_t), \mathbf{w}_t - \mathbf{z}_t \right\rangle \geq \bar{f}^{(t)}(\mathbf{w}_t) - \bar{f}^{(t)}(\mathbf{z}_t).
\end{aligned}$$

Therefore,

$$\sum_{t=1}^{T} v_t^2 \eta_t \mathbb{E}\left[ \bar{f}^{(t)}(\mathbf{w}_t) - \bar{f}^{(t)}(\mathbf{z}_t) \right] \leq \frac{1}{2} v_0^2 \|\mathbf{w}_1 - \mathbf{w}_\star\|^2 + \frac{1}{2} \sum_{t=1}^{T} v_t^2 \eta_t^2 \mathbb{E} \|\mathbf{g}_t\|^2.$$

On the other hand, $\mathbf{z}_t$ can be written directly as a convex combination of $\mathbf{w}_1, \ldots, \mathbf{w}_J$ and $\mathbf{w}_\star$, as follows:

$$\mathbf{z}_t = \frac{v_0}{v_t} \mathbf{w}_\star + \sum_{s=1}^{t} \frac{v_s - v_{s-1}}{v_t} \mathbf{w}_s.$$

Jensen's inequality then implies, using convexity of $\bar{f}^{(t)}$:

$$\sum_{t=1}^{J} v_t^2 \eta_t \mathbb{E}\left[\bar{f}^{(t)}(\mathbf{w}_t) - \bar{f}^{(t)}(\mathbf{z}_t)\right]$$

$$\geq \sum_{t=1}^{J} v_t^2 \eta_t \mathbb{E}\left[\bar{f}^{(t)}(\mathbf{w}_t) - \frac{v_0}{v_t} \bar{f}^{(t)}(\mathbf{w}_\star) - \sum_{s=1}^{t} \frac{v_s - v_{s-1}}{v_t} \bar{f}^{(t)}(\mathbf{w}_s)\right]$$

$$= \sum_{t=1}^{J} v_t \eta_t \mathbb{E}\left[v_t \bar{f}^{(t)}(\mathbf{w}_t) - v_0 \bar{f}^{(t)}(\mathbf{w}_\star) - \sum_{s=1}^{t} (v_s - v_{s-1}) \bar{f}^{(t)}(\mathbf{w}_s)\right]$$

$$= \sum_{t=1}^{J} v_t \eta_t \mathbb{E}\left[v_t \left(\bar{f}^{(t)}(\mathbf{w}_t) - \bar{f}^{(t)}(\mathbf{w}_\star)\right) - \sum_{s=1}^{t} (v_s - v_{s-1}) \left(\bar{f}^{(t)}(\mathbf{w}_s) - \bar{f}^{(t)}(\mathbf{w}_\star)\right)\right]$$

Combining the two bounds and denoting $\tilde{\delta}_t \triangleq \bar{f}^{(t)}(\mathbf{w}_t) - \bar{f}^{(t)}(\mathbf{w}_\star)$, we conclude that

$$\sum_{t=1}^{J} v_t \eta_t \mathbb{E}\left[v_t \tilde{\delta}_t - \sum_{s=1}^{t} (v_s - v_{s-1}) \left(\bar{f}^{(t)}(\mathbf{w}_s) - \bar{f}^{(t)}(\mathbf{w}_\star)\right)\right] \leq \frac{v_0^2}{2} \|\mathbf{w}_1 - \mathbf{w}_\star\|^2$$

$$+ \frac{1}{2} \sum_{t=1}^{J} v_t^2 \eta_t^2 \mathbb{E}\|\mathbf{g}_t\|^2.$$

Now, by assumption, for any $s \leq t, m \in [M]$:

$$\forall \mathbf{w}: \ f^{(t)}(\mathbf{w}; m) - f^{(t)}(\mathbf{w}_\star; m) \leq f(\mathbf{w}; m) - f(\mathbf{w}_\star; m) \leq (1+\eta_s \beta) \left(f^{(s)}(\mathbf{w}; m) - f^{(s)}(\mathbf{w}_\star; m)\right),$$

hence, taking expectations over $m \sim \tau$, we obtain (w.p. 1 w.r.t. randomness of $\mathbf{w}_s$);

$$-(1 + \eta_s \beta) \tilde{\delta}_s = -(1 + \eta_s \beta) \left(\bar{f}^{(s)}(\mathbf{w}_s) - \bar{f}^{(s)}(\mathbf{w}_\star)\right) \leq -\left(\bar{f}^{(t)}(\mathbf{w}_s) - \bar{f}^{(t)}(\mathbf{w}_\star)\right).$$

Combining with the previous display, we now have

$$\sum_{t=1}^{J} v_t \eta_t \mathbb{E}\left[v_t \tilde{\delta}_t - \sum_{s=1}^{t} (v_s - v_{s-1})(1 - \eta_s \beta) \tilde{\delta}_s\right] \leq \frac{v_0^2}{2} \|\mathbf{w}_1 - \mathbf{w}_\star\|^2 + \frac{1}{2} \sum_{t=1}^{J} v_t^2 \eta_t^2 \mathbb{E}\|\mathbf{g}_t\|^2,$$

which leads to the following after changing the order of summation;

$$\sum_{t=1}^{J} \left(\eta_t v_t^2 - (1 - \eta_t \beta)(v_t - v_{t-1}) \sum_{s=t}^{J} \eta_s v_s\right) \mathbb{E}\tilde{\delta}_t \leq \frac{v_0^2}{2} \|\mathbf{w}_1 - \mathbf{w}_\star\|^2 + \frac{1}{2} \sum_{t=1}^{J} v_t^2 \eta_t^2 \mathbb{E}\|\mathbf{g}_t\|^2,$$

and completes the proof. $\qquad \square$

Next, we prove a technical lemma which we employ in conjunction with the above in the proof of Lemma E.1.

**Lemma E.3.** *Let* $k \in \mathbb{N}$, $\beta > 0, a_1 > 0, a_2 > 0$, $\eta \in (0, \frac{3}{(8a_1+5a_2)\beta}]$, $\eta_t = \eta \cdot \frac{k-t+1}{k}$ *for* $t \in \{1, 2, \ldots, k\}$, $v_t = \frac{2}{k-t+1} + \frac{1}{k}$ *for* $t \in \{0, 1, \ldots, k-1\}$ *and* $v_k = v_{k-1} = 1 + \frac{1}{k}$. *Denote* $c_t = \eta_t v_t^2 - a_1 \beta \eta_t^2 v_t^2 - (1 + a_2\eta_t\beta)(v_t - v_{t-1})\sum_{s=t}^{k} \eta_s v_s$. *Then for all* $t \in \{1, 2, \ldots, k\}$, $c_t \geq 0$, *and in particular,* $c_k \geq \frac{\eta}{k}$.

*Proof.* As $v_k = v_{k-1}$ and $\eta \leq \frac{3}{(8a_1+5a_2)\beta}$,

$$c_k = \eta_k v_k^2 - a_1\beta\eta_k^2 v_k^2 = \eta_k v_k^2 \left(1 - \frac{a_1\beta\eta}{k}\right) = \frac{\eta}{k}\left(1 + \frac{1}{k}\right)^2 \left(1 - \frac{a_1\beta\eta}{k}\right)$$

$$\geq \frac{\eta}{k}\left(1 + \frac{1}{k}\right)\left(1 - \frac{3}{8k}\right) = \frac{\eta}{k}\left(1 + \frac{5}{8k} - \frac{3}{8k^2}\right) \geq \frac{\eta}{k}.$$

We proceed to lower bound $c_t$ for $t < k$. Focusing on the first terms, $A_t \triangleq \eta_t v_t^2 - a_1\beta\eta_t^2 v_t^2$,

$$A_t = \eta_t v_t^2 (1 - a_1\beta\eta_t) = \frac{\eta(k-t+1)}{k}\left(\frac{2}{k-t+1} + \frac{1}{k}\right)^2 (1 - a_1\beta\eta_t)$$

$$= \eta\left(\frac{4}{k(k-t+1)} + \frac{4}{k^2} + \frac{k-t+1}{k^3}\right)(1 - a_1\beta\eta_t)$$

$$\geq \eta\left(\frac{4}{k(k-t+1)} + \frac{4}{k^2}\right)(1 - a_1\beta\eta_t).$$

Moving to the last term, $B_t \triangleq (1 + a_2\beta\eta_t)(v_t - v_{t-1})\sum_{s=t}^{k} \eta_s v_s$,

$$B_t = (1 + a_2\beta\eta_t)\eta\left(\frac{2}{k-t+1} - \frac{2}{k-t+2}\right)\left(\frac{1 + \frac{1}{k}}{k} + \sum_{s=t}^{k-1}\left(\frac{2}{k} + \frac{k-s+1}{k^2}\right)\right)$$

$$= (1 + a_2\beta\eta_t)\frac{2\eta}{k(k-t+1)(k-t+2)}\left(1 + \frac{1}{k} + 2(k-t) + \frac{1}{k}\sum_{s=t}^{k-1}(k-s+1)\right)$$

$$= (1 + a_2\beta\eta_t)\frac{2\eta}{k(k-t+1)(k-t+2)}\left(1 + \frac{1}{k} + 2(k-t) + \frac{(k-t+3)(k-t)}{2k}\right)$$

$$= (1 + a_2\beta\eta_t)\frac{\eta(2k + 2 + 4k(k-t) + (k-t+3)(k-t))}{k^2(k-t+1)(k-t+2)}$$

$$= (1 + a_2\beta\eta_t)\eta\left(\frac{-6}{k(k-t+1)(k-t+2)} + \frac{4}{k(k-t+1)} + \frac{1}{k^2}\right)$$

$$\leq (1 + a_2\beta\eta_t)\eta\left(\frac{4}{k(k-t+1)} + \frac{1}{k^2}\right).$$

Thus, for $t < k$,

$$\frac{c_t}{\eta} \geq \left(\frac{4}{k(k-t+1)} + \frac{4}{k^2}\right)(1 - a_1\beta\eta_t) - (1 + a_2\beta\eta_t)\left(\frac{4}{k(k-t+1)} + \frac{1}{k^2}\right)$$

$$= \frac{3}{k^2} - \beta\eta_t\left(\frac{4a_1 + 4a_2}{k(k-t+1)} + \frac{4a_1 + a_2}{k^2}\right)$$

$$= \frac{3}{k^2} - \beta\eta\left(\frac{4a_1 + 4a_2}{k^2} + \frac{(4a_1 + a_2)(k-t+1)}{k^3}\right)$$

$$\geq \frac{3}{k^2} - \frac{\beta\eta}{k^2}(8a_1 + 5a_2).$$

Thus, for $\eta \leq \frac{3}{(8a_1+5a_2)\beta}$, $c_t \geq 0$. $\qquad \square$

The next lemma provides the stability property, which we leverage to translate our loss guarantees to the seen-task loss defined in Definition 4.8.

**Lemma E.4.** *Assume the conditions of Lemma E.1 and consider the algorithm defined in Eq.* (1) *with non-increasing step sizes* $\eta_t \leq 1/2\beta$. *In addition, define for every* $1 \leq k$, $\hat{f}_{1:k}(\mathbf{w}) \triangleq \frac{1}{k} \sum_{t=1}^{k} f(\mathbf{w}; \tau_t)$. *For all* $1 \leq k$, *the following holds:*

$$\mathbb{E}\hat{f}_{1:k}(\mathbf{w}_{k+1}) \leq 2\mathbb{E}f(\mathbf{w}_{k+1}) + \frac{8\beta^2\eta \|\mathbf{w}_1 - \mathbf{w}_\star\|^2}{k+1}.$$

*Proof.* First, any $\beta$-smooth $h : \mathbb{R}^d \to \mathbb{R}$ holds that

$$|h(\tilde{\mathbf{w}}) - h(\mathbf{w})| \leq \left|\nabla h(\mathbf{w})^\top (\tilde{\mathbf{w}} - \mathbf{w})\right| + \frac{\beta}{2} \|\tilde{\mathbf{w}} - \mathbf{w}\|^2$$
$$\leq \frac{1}{2\beta} \|\nabla h(\mathbf{w})\|^2 + \frac{\beta}{2} \|\tilde{\mathbf{w}} - \mathbf{w}\|^2 + \frac{\beta}{2} \|\tilde{\mathbf{w}} - \mathbf{w}\|^2 \qquad \text{(Young's ineq.)}$$
$$\leq h(\mathbf{w}) + \beta \|\tilde{\mathbf{w}} - \mathbf{w}\|^2.$$

Denote $f_m \triangleq f(\cdot; m)$ for all $m \in [M]$. Now, similarly the standard stability $\iff$ generalization argument [39, 19], and denoting by $\mathbf{w}_s^{(i)}$ the iterate after $s$ steps on the training set where the $i$'th example, $m_i$ was resampled (we denote the new example by $m_i'$):

$$\left|\mathbb{E}\left[f(\mathbf{w}_{k+1}) - \hat{f}_{1:k}(\mathbf{w}_{k+1})\right]\right| = \left|\frac{1}{k} \sum_{i=1}^{k} \mathbb{E}_{m_i \sim \tau} \left[f(\mathbf{w}_{k+1}; m_i) - f(\mathbf{w}_{k+1}^{(i)}; m_i)\right]\right|$$
$$\leq \frac{1}{k} \sum_{i=1}^{k} \mathbb{E}\left[f(\mathbf{w}_{k+1}; m_i) + \beta \left\|\mathbf{w}_{k+1}^{(i)} - \mathbf{w}_{k+1}\right\|^2\right]$$
$$= \mathbb{E}f(\mathbf{w}_{k+1}) + \frac{\beta}{k} \sum_{i=1}^{k} \mathbb{E}\left\|\mathbf{w}_{k+1}^{(i)} - \mathbf{w}_{k+1}\right\|^2.$$

Next, we bound $\left\|\mathbf{w}_{k+1}^{(i)} - \mathbf{w}_{k+1}\right\|^2$. Since by Lemma E.1, for every $t$, $f^{(t)}$ is convex and $\beta$-smooth, by the non-expansiveness of gradient steps in the convex and $\beta$-smooth regime when for every $t$, $\eta_t \leq 2/\beta$ [see Lemma 3.6 in 19]:

$$s \leq i \implies \left\|\mathbf{w}_s^{(i)} - \mathbf{w}_s\right\| = 0,$$
$$i < s \implies \left\|\mathbf{w}_{s+1}^{(i)} - \mathbf{w}_{s+1}\right\|^2 \leq \left\|\mathbf{w}_{i+1}^{(i)} - \mathbf{w}_{i+1}\right\|^2.$$

In addition, denoting by $f_{m_i'}$ the function that sampled after replacing $f_{m_i}$ and its corresponding time varying objective by $f^{(m_i')}$, by the conditions in Lemma E.1, we have that,

$$\left\|\mathbf{w}_{i+1}^{(i)} - \mathbf{w}_{i+1}\right\|^2 = \left\|\mathbf{w}_i^{(i)} - \eta_i \nabla f^{(m_i')}(\mathbf{w}_i^{(i)}) - \left(\mathbf{w}_i - \eta_i \nabla f^{(m_i)}(\mathbf{w}_i)\right)\right\|^2$$
$$= \eta_i^2 \left\|\nabla f^{(m_i')}(\mathbf{w}_i^{(i)}) - \nabla f^{(m_i)}(\mathbf{w}_i)\right\|^2$$
$$\leq 2\eta_i^2 \left\|\nabla f^{(m_i')}(\mathbf{w}_i^{(i)})\right\|^2 + 2\eta_i^2 \left\|\nabla f^{(m_i)}(\mathbf{w}_i)\right\|^2$$
$$\leq 4\beta\eta_i^2 f^{(m_i')}(\mathbf{w}_i^{(i)}) + 4\beta\eta_i^2 f^{(m_i)}(\mathbf{w}_i)$$
$$\leq 4\beta\eta_i^2 f_{m_i'}(\mathbf{w}_i^{(i)}) + 4\beta\eta_i^2 f_{m_i}(\mathbf{w}_i),$$

and, taking expectations,

$$\left\|\mathbf{w}_{i+1}^{(i)} - \mathbf{w}_{i+1}\right\|^2 \leq 8\beta\eta_i^2 \mathbb{E}f_{m_i}(\mathbf{w}_i),$$

Now,

$$\frac{\beta}{k} \sum_{i=1}^{k} \mathbb{E} \left\| \mathbf{w}_{k+1}^{(i)} - \mathbf{w}_{k+1} \right\|^2 \leq 8\beta^2 \, \mathbb{E} \left[ \frac{1}{k} \sum_{i=1}^{k} \eta_i^2 f_{m_i}(\mathbf{w}_i) \right]$$

$$\leq 8\beta \mathbb{E} \left[ \frac{1}{k} \sum_{i=1}^{k} \eta_i f_{m_i}(\mathbf{w}_i) \right].$$

Summarizing, we have shown that:

$$\left| \mathbb{E} \left[ f(\mathbf{w}_{k+1}) - \hat{f}_{1:k}(\mathbf{w}_{k+1}) \right] \right| \leq \mathbb{E} f(\mathbf{w}_{k+1}) + \frac{\beta}{k} \sum_{i=1}^{k} \mathbb{E} \left\| \mathbf{w}_{k+1}(i) - \mathbf{w}_{k+1} \right\|^2$$

$$\leq \mathbb{E} f(\mathbf{w}_{k+1}) + 8\beta \mathbb{E} \left[ \frac{1}{k} \sum_{i=1}^{k} \eta_i f_{m_i}(\mathbf{w}_i) \right].$$

Now, by Theorem E.2 with $v_t = 1$ for every $t$, we have, since $\eta_t \beta \leq \frac{1}{4}$, $\frac{1}{1+\eta_t \beta} \geq \frac{4}{5}$

$$\frac{4}{5} \sum_{i=1}^{k} \eta_i \mathbb{E} f_{m_i}(\mathbf{w}_i) = \frac{4}{5} \sum_{i=1}^{k} \eta_i \mathbb{E} f(\mathbf{w}_i) \qquad\qquad (\mathbb{E} f_{m_i}(w_i) = \mathbb{E} f(w_i))$$

$$\leq \sum_{i=1}^{k} \eta_i \mathbb{E} \bar{f}^{(i)}(\mathbf{w}_i)$$

$$= \sum_{i=1}^{k} \eta_i \mathbb{E} \left[ \bar{f}^{(i)}(\mathbf{w}_i) - \bar{f}^{(i)}(\mathbf{w}_\star) \right]$$

$$\leq \frac{1}{2} \left\| \mathbf{w}_1 - \mathbf{w}_\star \right\|^2 + \frac{1}{2} \sum_{i=1}^{k} \eta_i^2 \mathbb{E} \left\| \nabla f^{(i)}(w_i) \right\|^2$$

$$\leq \frac{1}{2} \left\| \mathbf{w}_1 - \mathbf{w}_\star \right\|^2 + \sum_{i=1}^{k} \beta \eta_i^2 \mathbb{E} f^{(i)}(w_i)$$

$$\leq \frac{1}{2} \left\| \mathbf{w}_1 - \mathbf{w}_\star \right\|^2 + \frac{1}{4} \sum_{i=1}^{k} \eta_i \mathbb{E} f_{m_i}(w_i),$$

this implies,

$$\sum_{i=1}^{k} \eta_i \mathbb{E} f_{m_i}(\mathbf{w}_i) \leq \left\| \mathbf{w}_1 - \mathbf{w}_\star \right\|^2.$$

Then we can conclude,

$$\left| \mathbb{E} \left[ f(\mathbf{w}_{k+1}) - \hat{f}_{1:k}(\mathbf{w}_k) \right] \right| \leq \mathbb{E} f(\mathbf{w}_{k+1}) + \frac{8\beta \left\| \mathbf{w}_1 - \mathbf{w}_\star \right\|^2}{k},$$

and the result follows. $\qquad\qquad\qquad\qquad\qquad\qquad\qquad\qquad\qquad\qquad\qquad\qquad\qquad$ $\square$

We are now ready to prove our main lemma for this section.

*Proof of Lemma E.1.* To begin, note that we are after a guarantee for $\mathbf{w}_{k+1}$, which is the SGD iterate that was produced by taking $k$ steps over $k$ losses. To that end, we are going to apply Theorem E.2 with $J = k + 1$, hence we are obligated to supply a random ordering $\tau \colon [k+1] \to [M]$, $f^{(k+1)}$ and $\eta_{k+1}$, which are not supplied in the statement of our lemma. Therefore, we define

$$\forall m \in [M] : f^{(k+1)}(\cdot;m) \triangleq f(\cdot;m), \text{ and } \eta_{k+1} \triangleq \eta\left(\frac{1}{k+1}\right) = \eta\left(\frac{(k+1)-(k+1)+1}{k+1}\right).$$

We additionally define $\bar{f}^{(k+1)}(\mathbf{w}) \triangleq \mathbb{E}_{m\sim\text{Unif}[\text{M}]} f^{(k+1)}(\mathbf{w};m)$. It is immediate to verify $f^{(k+1)}$ satisfies the properties required from $f^{(t)}$ for $t \in [k]$ and $\eta_{k+1}$ is the next step size in the sequence $\eta_1, \ldots, \eta_k$ defined in the statement. Finally, we simply define the extra sampled index $\tau_{k+1}$ to be uniform over $[M]$, exactly like $\tau_t$ for $t \in [k]$.

Now, the conditions for Theorem E.2 are immediately satisfied with $J = k + 1$ by our assumptions and augmentation described above, leading to:

$$\sum_{t=1}^{k+1} \left(\eta_t v_t^2 - (1 - \eta_t\beta)(v_t - v_{t-1}) \sum_{s=t}^{k+1} \eta_s v_s\right) \mathbb{E}\left[\bar{f}^{(t)}(\mathbf{w}_t) - \bar{f}^{(t)}(\mathbf{w}_\star)\right]$$

$$\leq \frac{v_0^2}{2} \|\mathbf{w}_1 - \mathbf{w}_\star\|^2 + \frac{1}{2} \sum_{t=1}^{k+1} \eta_t^2 v_t^2 \mathbb{E}\left\|\nabla f^{(t)}(\mathbf{w}_t;\tau_t)\right\|^2.$$

Now, by the joint realizability assumption, conditioning on all randomness up to round $t$,

$$\mathbb{E}_{\tau_t}\left\|\nabla f^{(t)}(\mathbf{w}_t;\tau_t)\right\|^2 \leq 2\beta\mathbb{E}_{\tau_t}[f^{(t)}(\mathbf{w}_t;\tau_t) - f^{(t)}(\mathbf{w}_\star;\tau_t)] = 2\beta\left(\bar{f}^{(t)}(\mathbf{w}_t) - \bar{f}^{(t)}(\mathbf{w}_\star)\right).$$

Combining with the previous display and rearranging, this yields

$$\sum_{t=1}^{k+1} \left(\eta_t v_t^2 - \beta\eta_t^2 v_t^2 - (1 - \eta_t\beta)(v_t - v_{t-1}) \sum_{s=t}^{k+1} \eta_s v_s\right) \mathbb{E}\left[\bar{f}^{(t)}(\mathbf{w}_t) - \bar{f}^{(t)}(\mathbf{w}_\star)\right]$$

$$\leq \frac{v_0^2}{2} \|\mathbf{w}_1 - \mathbf{w}_\star\|^2. \quad (2)$$

Now, by Lemma E.3, the step size sequence $\eta_t = \eta(\frac{(k+1)-t+1}{k+1})$ with $\eta \leq \frac{3}{13\beta}$ and $\{v_t\}_{t=1}^{k+1}$ as specified by the lemma, guarantee that $c_{k+1} \geq \frac{\eta}{k+1}$, $v_0 \leq 3/(k+1)$, and $c_t \geq 0$ for all $t \in [k+1]$. Combining these properties with Eq. (2) we obtain,

$$\mathbb{E}\left[f(\mathbf{w}_{k+1}) - f(\mathbf{w}_\star)\right] = \mathbb{E}\left[\bar{f}^{(k+1)}(\mathbf{w}_{k+1}) - \bar{f}^{(k+1)}(\mathbf{w}_\star)\right]$$

$$\leq \frac{v_0^2}{2c_{k+1}} \|\mathbf{w}_1 - \mathbf{w}_\star\|^2 \leq \frac{9}{2\eta(k+1)} \|\mathbf{w}_1 - \mathbf{w}_\star\|^2,$$

which completes the proof for the first part. For the seen-task guarantee, by Lemma E.4, we have

$$\mathbb{E}\left[\frac{1}{k} \sum_{t=1}^{k} f(\mathbf{w}_{k+1};\tau_t) - f(\mathbf{w}_\star;\tau_t)\right] \leq 2\mathbb{E}f(\mathbf{w}_{k+1}) + \frac{8\beta^2\eta \|\mathbf{w}_1 - \mathbf{w}_\star\|^2}{k+1},$$

which gives, after combining with the population loss guarantee:

$$\mathbb{E}\left[\frac{1}{k} \sum_{t=1}^{k} f(\mathbf{w}_{k+1};\tau_t) - f(\mathbf{w}_\star;\tau_t)\right] \leq \frac{18}{\eta(k+1)} \|\mathbf{w}_1 - \mathbf{w}_\star\|^2 + \frac{8\beta^2\eta \|\mathbf{w}_1 - \mathbf{w}_\star\|^2}{k+1}$$

$$\leq \frac{20 \|\mathbf{w}_1 - \mathbf{w}_\star\|^2}{\eta(k+1)}, \quad\quad (\eta \leq 1/(2\beta))$$

which completes the proof. □

