# OpenReview forum: "Optimal Rates in Continual Linear Regression via Increasing Regularization"
_NeurIPS.cc/2025/Conference — NeurIPS 2025 poster_

### Official Review · Reviewer_ahtS · 2025-07-02

**Clarity:** 3
**Significance:** 4
**Originality:** 4
**Rating:** 5
**Confidence:** 2

**Summary:**

This paper derives two convergence rates for continual linear regression with regularization: for fixed regularization, they achieve $O(log k/k)$ convergence rate, whereas for scheduled regularization they achieve $O(1/k)$. As $O(1/k)$ is the optimal rate one can achieve for continual learning, they have achieved optimal convergence rate for continual learning for the first time. The proof is followed by first understanding the problem as Incremental Gradient Descent, and then applying the bound in [1].

[1] Evron, Itay, et al. "Better rates for random task orderings in continual linear models." arXiv preprint arXiv:2504.04579 (2025).

**Questions:**

A general question I have is "is joint realizability a reasonable assumption to study continual learning?" I get that for continual learning, both learning from new tasks and remembering previous tasks are important. To me joint realizability feels like the framework doesn't have the notion of forgetting - because naively trying to do better for every given task naturally leads to better performance for all past tasks. Are there different setups in continual learning theory that can reflect such issues? Is your theory applicable in those cases too?

**Ethical Concerns:**

["NO or VERY MINOR ethics concerns only"]

**Final Justification:**

There was basically no issue in this paper: the contributions were solid and to me they could potentially be very influential in continual learning community. A question I had was is the joint realization assumption reasonable, and the authors pointed out to an interesting phenomena which shows that it is indeed the case. So I did not change my initial assessment.

**Limitations:**

One limitation is that the result is purely theoretical.

**Quality:**

3

**Strengths And Weaknesses:**

Strengths: A novel scheme that has optimal convergence rate is very valuable. The fact that continual learning can be understood as incremental gradient descent is also important I believe, as it enables continual learning to be studied by tools from SGD. They also consider many diverse settings, such as considering seen-task loss, different regularizations, etc.

Weaknesses: One weakness I find is that there is no experimental results. I understand that this is a purely theoretical paper: of course showing that increasing regularization strength for isotropic regularization would have been ideal, but at least showing that the lower bounds / upper bounds make sense for certain continual linear regression problems could have been good.

---

> ### Author Rebuttal · Authors · 2025-07-31
>
> We thank the reviewer for their positive feedback. We are happy that they find our schedules **novel** and **“very valuable”**, and that they appreciate the SGD perspective that we derive.
>
> We like to highlight that—as the review summary mentions—we did employ the bound from Evron et al. [13] while proving the nearly-optimal rate for fixed regularization strengths (Lemma 4.3); however, the proof of the optimal rate for increasing strengths required novel analytic techniques (Lemma 4.7), not covered by the previous analysis of [13].
>
> The reviewer mentioned as a **weakness** that we do not have **experimental results**:
> As the reviewer mentioned, this is a purely theoretical paper focused on establishing convergence guarantees. That said, we agree that simple experiments can help illustrate the behavior of our regularization schedules and highlight the practical implications of our theoretical results. We will work on such experiments, but due to the short rebuttal period (which doesn't allow any figures), we will only be able to do so for the final version.
>
> ---
>
> We now answer the reviewer's important **question** about the *joint realizability assumption*:
>
> - First, please note that we assume joint realizability solely for the training data. This assumption is quite **common in theoretical work** [e.g., 11,12,13,16,25,35] and is **practically reasonable in overparameterized models**. In particular, it was shown empirically [\*1] and theoretically [\*2; Theorem 1] that a sufficiently overparameterized neural network can perfectly fit data, even with random labels.
> This holds also to linearized approximations of such networks (e.g., using the Neural Tangent Kernel).
>
> - The reviewer also mentioned that “joint realizability feels like the framework doesn't have the notion of forgetting”. However, this intuition is misleading. For example, a prior prominent work (Evron et al. [11]) illustrated geometrically how even under joint realizability, solving a task would often lead to forgetting the previous one (i.e., leaving its solution space; see their Figure 1).
> On the other hand, they showed that learning any single task 'contracts' the learner toward the offline (joint) solution, agreeing more with the reviewer's intuition.
> Even so, they presented adversarial task sequences under which the learner learns *virtually nothing*—i.e., forgets everything—even after observing an *infinitely long sequence of jointly realizable tasks*.
>
> - Moreover, like our Table 1 shows, prior work has proved that the lower and upper bounds on the convergence of continual learning schemes may differ (quite significantly), even under the joint realizability assumption. Our work shows that regularization, widely used in practice, theoretically achieves optimal rates under that assumption.
>
> We will use the discussion here to illuminate these aspects in the revised manuscript.
>
> ---
>
> **Additional references:**
> [\*1] C. Zhang, S. Bengio, M. Hardt, B. Recht, and O. Vinyals. Understanding deep learning requires rethinking generalization. In ICLR, 2017.
> [\*2] E. B. Baum. On the capabilities of multilayer perceptrons. Journal of Complexity, 1988.

---

> > ### Comment · Reviewer_ahtS · 2025-08-05
> >
> > Thank you for the thorough response. It is interesting that there exists an adversarial task sequence where joint realizability cannot help the model improve consistently, and frankly the opposite, make the model forget everything. I see the value of the paper much more clearly now. I think my initial assessment was fair, and I keep my score.

---

### Official Review · Reviewer_6cuU · 2025-07-03

**Clarity:** 3
**Significance:** 3
**Originality:** 3
**Rating:** 4
**Confidence:** 3

**Summary:**

This paper addresses the problem of realizable continual linear regression under random task orderings, a setting that is important for understanding and advancing continual learning. Previous work has established only suboptimal learning rates, achieving an upper bound of \$O(1/k^{1/4})\$ despite the existence of a lower bound of \$\Omega(1/k)\$. The authors show that incorporating regularization—either explicit L2 regularization or implicit regularization via a finite number of gradient steps—significantly improves convergence. By reducing these schemes to stochastic gradient descent (SGD) on appropriately defined surrogate losses, they demonstrate that a fixed regularization strength leads to a near-optimal rate of \$O(\log k / k)\$, while an increasing regularization schedule achieves the optimal rate of \$O(1/k)\$. These results close the theoretical gap and highlight the practical and theoretical importance of tuning regularization in continual learning.

**Questions:**

See Weaknesses above

**Ethical Concerns:**

["NO or VERY MINOR ethics concerns only"]

**Final Justification:**

I recommend accepting this paper because it tries to tackle a fundamental theoretical problem, and makes notable advances.

**Limitations:**

See Weaknesses above

**Quality:**

3

**Strengths And Weaknesses:**

**Strengths:**

1. The paper tackles an important and fundamental problem in continual learning—the learning bounds of continual linear regression.

2. The writing is clear, well-structured, and easy to follow.

3. The theoretical results appear sound and provide tighter bounds compared to existing work.

**Weaknesses:**

1. The primary concern lies in the parameter setting of the algorithm. The data radius \$R\$ must be known in advance to determine the regularization coefficient. Is this assumption realistic in practical scenarios?

2. In Figure 1, the loss and forgetting rate are combined into a single quantity. However, continual learning is typically evaluated using two separate metrics: average accuracy across all tasks and forgetting rate on previous tasks. Why do the authors merge these two metrics into one?

3. Could the authors include synthetic experiments to further validate the effectiveness of the proposed method?

---

> ### Author Rebuttal · Authors · 2025-07-31
>
> We thank the reviewer for their positive feedback. We are happy that they find our paper **clear**, providing **sound results** on a **fundamental problem**.
>
> We wish to resolve the reviewer’s “primary concern” (1) and clarify the analyzed metrics (2).
> 1. **Knowledge of the data radius R:**
> The required knowledge of problem parameters for optimal tuning is a common issue in virtually *any* stochastic optimization method, perhaps most notoriously in SGD, where the choice of step size crucially depends on unknown parameters, such as the Lipschitz constant and the initial distance from a minimizer (see, e.g., elaborate discussion in [\*1]).
> In our continual learning case, the schedules require knowledge only about the data radius—a quantity that could be easily inferred from unlabeled training data in many real-world problems.
> 1. **“Single” loss metric in Figure 1:**
> The reviewer is correct in noting that loss and forgetting are distinct quantities. However, in the random orderings we study, these quantities are closely related—as explicitly discussed in Remark 2.2. While most of the paper focuses on the *loss* convergence, we discuss forgetting in length in Section 4.3 (titled: *“Do not forget forgetting: Extension to seen-task loss”*) and explain that the optimal rate we derived extends to it as well.
>
> Furthermore, regarding the reviewer’s third remark (3),
>
> 3. **Synthetic Experiments:**
> Our paper is theoretical, focused on establishing formal convergence guarantees in continual linear regression. That said, we agree that simple illustrative experiments can help clarify the behavior of our regularization schedules and support the intuition behind our results.
> We will work on such experiments, but due to the short rebuttal period (which doesn't allow any figures), we will only be able to do so for the final version.
>
>
> We will be happy to answer any additional questions during the discussion period.
>
> ---
> **Additional references:**
> [*1] Y. Carmon and O. Hinder. The price of adaptivity in stochastic convex optimization. COLT 2024.

---

> > ### Comment · Reviewer_6cuU · 2025-08-03
> >
> > I thank the authors for their response and clarification, and I have decided to keep my score.

---

### Official Review · Reviewer_7nbE · 2025-07-03

**Clarity:** 2
**Significance:** 2
**Originality:** 3
**Rating:** 4
**Confidence:** 2

**Summary:**

This paper studies continual linear regression under random task orderings, wherein the learner faces a collection of randomly-ordered linear regression tasks and seeks to minimize the average error across them. A $\log{k}/k$ last-iterate convergence rate is derived using a fixed regularizer, and the result is improved to a $1/k$ rate using increasing regularization; both results significantly improve the best-known rate of $1/k^{1/4}$. A lower bound of $1/k$ is also shown, demonstrating that the $1/k$ rate is optimal.

**Questions:**

- The comment before Lemma 4.9 was unclear to me; how can we simultaneously have a 1/k lower bound and non-isotropic regularization ensuring seen-task loss of zero? The elaboration of this comment in Section 6 did not resolve this question for me either.

- I don't really understand the point of presenting both the fixed and increasing regularization scheme; the former seems to be strictly worse, so why not just present the optimal algorithm? It feels like this could have been included in the appendix rather than being featured prominently in the main text.

**Ethical Concerns:**

["NO or VERY MINOR ethics concerns only"]

**Final Justification:**

The paper has some novel results, but relatively weak writing which is left unaddressed by unconvincing justifications in the rebuttal. After reading the response and the other reviews, I've retained my original score.

**Limitations:**

yes

**Quality:**

3

**Strengths And Weaknesses:**

## Strengths

There are two common approaches to the problem studied: applying explicit regularization on each step (to penalize moving too far from the current iterate, thus "forgetting" much about previous tasks), and implicitly regularizing by applying only a limited amount of updates in each task. Section three shows that both of these options can be analyzed in terms of SGD on a particular surrogate losses, leading to a more unified analysis. Very clean.

## Weaknesses

- The lower bound is not particularly novel; it is standard but technically hasn't been explicitly stated in the literature (the paper is up front about this)

- The writing and paper structure is not very polished. Generally the writing comes across as rushed.
  -  e.g. Section 2 introduces the setting but only partially, with key assumptions (realizability and random task ordering) not being introduced until later, in Section 4.
  - Section 4 introduces the 1/k lower bound, and then follows up with an algorithm which fails to achieve it, only to fix it with a separate algorithm afterwards. It seems redundant to present the sub-optimal algorithm if an optimal one is obtained at seemingly no compromise.
  - Lemma 4.7 re-introduces all the assumptions in the lemma statement. This one lemma take up more than a third of a page

---

> ### Author Rebuttal · Authors · 2025-07-30
>
> We thank the reviewer for their positive feedback. We are happy that they find our analysis clean and that they appreciate how we **unified two common continual learning practical algorithms** under an analytical framework that has ties to the rich literature on SGD.
>
> As **weaknesses**, the reviewer mostly pointed out **places where the paper could be clearer**. We would like to clarify these issues here—and later in our manuscript’s next version:
> 1. **Lower bound is “not particularly novel” (but it is also not presented as a main contribution).**
> This bound is indeed standard (and like the reviewer acknowledged, we were explicit about it).
> However, we do not consider it a 'weakness' of our paper, as it was not presented as a primary contribution but rather included for completeness to facilitate the discussion of the optimality of our novel *upper* bounds.
>
>
> 1. **Reservations regarding the paper’s structure.**
> We gave careful consideration to the appropriate structure—and particularly to the issues that the reviewer raised. Below, we explain our rationale. Still, following the reviewer’s remarks we will try to improve the flow—especially when transitioning between sections and/or around key results.
>     - **Setting “partially” introduced in Section 2.**
> While the reviewer proposed a valid alternative, we believe that our structure offers certain advantages: the reductions in Section 3 apply **without assuming realizability or random task ordering**. By introducing those assumptions only later—in Section 4—we kept Section 3 broadly applicable. For example, future work on adversarial/structured task sequences can use our reductions directly without modification.
>     - **Length and placement of Lemma 4.7.**
> Lemma 4.7 offers a result of independent interest: the **first optimal last-iterate convergence rate for SGD** in a realizable setting. To make it accessible to readers from an SGD/optimization background, we chose to state it fully, in a self-contained version that is more *general* than the rest of the paper (e.g., it applies to all smooth and convex functions).
> We will clarify this when presenting the lemma.
>
>
> Regarding the reviewer’s **questions**:
> 1. **Lemma 4.9’s preamble (isotropic vs non-isotropic):**
> Please note that Lemma 4.9 states a lower bound for Scheme 1 which employs *isotropic* regularization. In Section 6 (lines 282-288) we discuss how non-isotropic regularization (requiring additional $d^2$ memory) achieves a seen-task loss of $0$.
> There is no contradiction between the two claims, because they deal with distinct regularization schemes. This is also what we tried to convey in Lemma 4.9’s preamble.
> Following the reviewer’s question, we will ensure that the word 'isotropic' explicitly accompanies 'Scheme 1' within Lemma 4.9 itself, and not just in its preamble.
>
>
> 1. **Why present the fixed-regularization scheme?**
> Fixed regularization is widely used in practice and merits theoretical analysis in its own right. Furthermore, our fixed-schedule results **extend to random task orderings without replacement** (see Remark 4.5), which the increasing schedule does not currently cover.
> We will emphasize this in Section 4.1.
>
> We again thank the reviewer for their thorough feedback, which we have used to improve the clarity of our paper.

---

> > ### Comment · Reviewer_7nbE · 2025-08-01
> >
> > I see; thank you for the thorough response!

---

### Official Review · Reviewer_ZBRc · 2025-07-03

**Clarity:** 3
**Significance:** 2
**Originality:** 3
**Rating:** 4
**Confidence:** 1

**Summary:**

This paper proves new worst-case bounds on linear regression problems in a continual learning setting, specifically focusing on the regularized case.

I should note that this paper is far out of my field of expertise, and thus my assessment is just an educated guess. I hope the other reviewers are more qualified to review the paper.

**Questions:**

- As someone working on other varieties of ML, such as large language models, what is the significance of this work?

**Ethical Concerns:**

["NO or VERY MINOR ethics concerns only"]

**Final Justification:**

I would like to follow the prevailing sentiment about this paper, as I am not very qualified to review it myself.

**Limitations:**

yes

**Paper Formatting Concerns:**

Small comment:
> Optimal rates via increasing regularization regularization
Repeats "regularization" twice

**Quality:**

3

**Strengths And Weaknesses:**

Strengths:
- Clarity: The paper is dense, but to me it seemed clearly written.
- Originality: Analyzing learning dynamics under regularization seems like an important topic to be covering, and if this has indeed not been covered in previous work then I think it is worth examining.

Weaknesses:
- Significance: I mostly work in more applied varieties of machine learning, and to me it was not very clear what the significance of this work would be. (Somewhat facetiously but also somewhat not,) we are approaching AGI and it's not clear that knowing this tighter bound will get us closer to that (although purchasing more GPUs likely will). It would be nice to have a little more grounding in what the practical implications of the work would be, and how they might eventually lead into new optimization algorithms, etc. That being said, this is not the variety of paper I typically read in the first place, so it might be that this is much more interesting to people for whom this is a central research interest.

---

> ### Author Rebuttal · Authors · 2025-07-30
>
> We thank the reviewer for their feedback. We are happy that they find our paper **clear** and **original** within the scope of theoretical continual learning.
>
> The reviewer mostly doubted the significance of an analytical work like ours—somewhat calling for a broader discussion on the importance of theoretical ML research while the community seems to be *“approaching AGI”*. While we do not feel this is the most appropriate platform for such a broad discussion, we would like to note that:
>
> 1. **Generally**, the rapid progress towards AGI might “hit a wall” which will require a better theoretical understanding to cross—perhaps in the sense that optical theory has helped develop better telescopes (which were first designed empirically, without theory). Better theory in ML should eventually be able to provide scaling laws, help devise better algorithms, and also explain the behavior of existing algorithms.
>
> 1. **Relevance to LLMs.**
> Continual learning is considered essential for general-purpose AI systems \[\*1 below]. Recent work \[e.g., \*2, \*3] shows that fine-tuning LLMs often leads to catastrophic forgetting, especially during domain adaptation.
> These works already use quadratic regularization like the one we study. While our results apply to linear models, they offer theoretically-grounded rule-of-thumbs into regularization schedules which may be useful in practical continual learning systems, perhaps especially in large models.
>
> 1. **Our work is important to the theoretical continual learning community.**
> As our Table 1 shows, our rate directly improves upon and/or relates to prior work from recent important conferences: Evron et al. (COLT 22), Kong et al. (NeurIPS 23), and Cai and Diakonikolas (ICLR 25). More broadly, the continual **linear** regression setting is a central setting, studied theoretically in many papers from notable conferences (e.g., [10, 11, 30, 15, 35, 27, 46, 16] cited in Line 58). Furthermore, the reviewer’s summary seemingly ignores a central, novel contribution of our paper: our work not only “proves new worst-case bounds in a continual learning setting”, but also shows how to achieve these optimal rates using explicit and implicit regularization.
>
> We respectfully ask the reviewer to reconsider their assessment of the significance of our work, taking into account the standards and interests of the theoretical continual learning community.
>
> ---
> Additional references:
> \[\*1] Hadsell, R., Rao, D., Rusu, A. A., & Pascanu, R. (2020). Embracing change: Continual learning in deep neural networks. Trends in Cognitive Sciences, 24(12), 1028–1040.
> \[\*2] Song, S. et al. (2025). How to alleviate catastrophic forgetting in LLM fine-tuning: Hierarchical layer-wise and element-wise regularization. arXiv preprint 2501.13669.
> \[\*3] Xiang, J. et al. (2023). Language models meet world models: Embodied experiences enhance language models. NeurIPS 2023.

---

> > ### Comment · Reviewer_ZBRc · 2025-08-01
> > **Thank you for the response**
> >
> > Thank you for the response! As I noted in my review, this is not my area of expertise. I have also checked the other reviews and they seem to be uniformly positive for the paper, so I will also raise my review score by one because my biggest question (that this may or may not be of interest to experts in the field) seems to fall on the side of domain experts being interested.

---

### Decision · Program_Chairs · 2025-09-17

**Decision:**

Accept (poster)

**Comment:**

The paper studies overparameterized linear regression in the continual learning setting with repeated tasks, under random task ordering. It is motivated by a gap left from prior work: the lower bound on the convergence rate was $\Omega(1/k)$, while the upper bound was $O(1/k^{1/4})$. This paper closes this gap. The main idea is in using increasing regularization strength between the task adaptation procedures, either via explicit regularization (i.e., by adding a quadratic term to the objective) or via implicit regularization (early stopping). All reviewers were in agreement that the paper presents solid theoretical contributions and is a good fit for NeurIPS.